# Latitudinal gradient in dairy production with the introduction of farming in Atlantic Europe

Miriam Cubas [1,2,3]✉, Alexandre Lucquin [1], Harry K. Robson[1], André Carlo Colonese[1,4,5], Pablo Arias[6], Bruno Aubry[7], Cyrille Billard[8], Denis Jan[9], Mariana Diniz[10], Ricardo Fernandes[11,12,13], Ramón Fábregas Valcarce [14], Cécile Germain-Vallée[9], Laurent Juhel[15], Arturo de Lombera-Hermida[14], Cyril Marcigny [16], Sylvain Mazet [7], Grégor Marchand [17], César Neves [10], Roberto Ontañón-Peredo [18], Xose Pedro Rodríguez-Álvarez[19,20], Teresa Simões[21], João Zilhão [10,22,23] & Oliver E. Craig [1]

The introduction of farming had far-reaching impacts on health, social structure and demography. Although the spread of domesticated plants and animals has been extensively tracked, it is unclear how these nascent economies developed within different environmental and cultural settings. Using molecular and isotopic analysis of lipids from pottery, here we investigate the foods prepared by the earliest farming communities of the European Atlantic seaboard. Surprisingly, we find an absence of aquatic foods, including in ceramics from coastal sites, except in the Western Baltic where this tradition continued from indigenous ceramic using hunter-gatherer-fishers. The frequency of dairy products in pottery increased as farming was progressively introduced along a northerly latitudinal gradient. This finding implies that early farming communities needed time to adapt their economic practices before expanding into more northerly areas. Latitudinal differences in the scale of dairy production might also have influenced the evolution of adult lactase persistence across Europe.

[1] BioArCh, Department of Archaeology, University of York, Wentworth Way, Heslington, York YO10 5DD, UK. [2] Departamento de Historia, Universidad de Oviedo, C/Amparo Pedregal s/n, E-33011 Oviedo, Spain. [3] Sociedad de Ciencias Aranzadi, Zorroagagaina 11, E-20014 Donostia-San Sebastian, Spain. [4] Department of Prehistory, Edifici B Facultat de Filosofia i Lletres, Universitat Autònoma de Barcelona, Carrer de la Fortuna, Bellaterra, E-08193 Barcelona, Spain. [5] Institute of Environmental Science and Technology (ICTA), Edifici Z, Universitat Autònoma de Barcelona, Carrer de les columns, Bellaterra, E-08193 Barcelona, Spain. [6] Instituto Internacional de Investigaciones Prehistóricas de Cantabria, Universidad de Cantabria-Gobierno de Cantabria, Avd de los Castros s/n, E-39005 Santander, Spain. [7] INRAP Centre Archéologique du Grand Quevilly, 30 boulevard de Verdun, 76120 Le Grand Quevilly, France. [8] DRAC du Département Normandie Service Régional de l'Archéologie, 13 bis, rue Saint-Ouen, 14052 CAEN cedex 4, France. [9] Service Archéologie du Conseil Départemental du Calvados, 36 rue Fred Scamaroni, 14000 Caen, France. [10] Centro de Arqueologia da Universidade de Lisboa -UNIARQ- Alameda da Universidade, 1600-214 Lisbon, Portugal. [11] Department of Archaeology, Max Planck Institute for the Science of Human History, 07745 Jena, Germany. [12] School of Archaeology, University of Oxford, 1 South Parks Road, Oxford OX1 3TG, UK. [13] Faculty of Arts, Masaryk University, Arne Nováka 1, 602 00 Brno-střed, Czech Republic. [14] Facultad de Geografía e Historia, University of Santiago de Compostela, Praza Universidade, 1, E-15782 Santiago de Compostela, Spain. [15] INRAP, Centre Archéologique de Cesson-Sévigné, 37 rue du Bignon CS 67737, 35577 Cesson-Sévigné cedex, France. [16] INRAP, Centre Archéologique de Bourguébus, Boulevard de l'Europe, 14540 Bourguébus, France. [17] Centre de Recherche en Archéologie Archéosciences Histoire, UMR 6566 CNRS - CReAAH, Campus Beaulieu - Bât 24 - 25. 263 avenue du Général Leclerc - CS 74 205, 35042 RENNES Cedex, France. [18] Museo de Prehistoria y Arqueología de Cantabria y Cuevas Prehistóricas de Cantabria-Instituto Internacional de Investigaciones Prehistóricas de Cantabria, Ruiz de Alda, 19, E-39009 Santander, Spain. [19] IPHES, Institut Català de Paleoecologia Humana i Evolució Social, C/ Marcel.lí Domingo s/n- Campus Sescelades URV (Edifici W3), E-43007 Tarragona, Spain. [20] Área de Prehistoria, Universitat Rovira i Virgili (URV), Avinguda de Catalunya 35, E-43002 Tarragona, Spain. [21] Museu Arqueológico de São Miguel de Odrinhas. Av. Prof. Dr. D. Fernando de Almeida, São Miguel de Odrinhas, São João das Lampas, 2705-739 Sintra, Portugal. [22] SERP (Seminari d'Estudis i Recerques Prehistòriques; SGR2017-00011), Departament d'Història i Arqueologia, Facultat de Geografia i Història, Universitat de Barcelona, c/ Montalegre 6, E-08001 Barcelona, Spain. [23] ICREA, Catalan Institution for Research and Advanced Studies, Passeig Lluís Companys 23, E-08010 Barcelona, Spain. ✉email: miriam.cubas@york.ac.uk

The motivations for the introduction of farming in Europe and the nature of the earliest farming communities are key topics in European prehistory. Traditionally this issue has been often reduced to polarised hypotheses of demic diffusion versus acculturation to describe processes applicable to the whole of Europe. More recently archaeologists have stressed the need to consider regional variations as well as putting forward other more complex models, e.g. refs. [1–3]. Our understanding of agricultural origins in Europe has also been reinvigorated by DNA analysis of human remains. These studies broadly support a Near Eastern origin for Europe's earliest farming communities directly associated with the beginning of food production, e.g. refs. [4–6]. Genetic studies, however, offer little detail regarding the social or economic drivers that led to the inception of the Neolithic, and currently there are insufficient data to track demographic change throughout all regions. Consequently, the regional development of early agro-pastoral economies is less well understood. Indeed, economic transformation may well have been independent of demographic changes and correlated instead to geographic and climatic factors as well as the types of interactions pioneer farmers had with various indigenous hunter–gatherer–fishers as they moved in and around their territories.

The widespread application of scientific methods, such as the stable isotope analysis of human remains and organic residue analysis of pottery, is beginning to highlight the varied nature of Europe's earliest farming communities. A surprising finding is that dairy production, once thought to have developed much later in the Neolithic[7], was a component of even the earliest Neolithic economies and may even have been one of the motivations for ruminant domestication[8]. Although milk derived lipids have now been identified in Early Neolithic pottery from Southern[9,10], Central[11–14] and Northern Europe[15–18], the scale and intensity of dairying in relation to meat production is still unknown. Indeed, cumulatively such studies are beginning to highlight regional patterns of variation in early animal husbandry[9,10,12,19], providing a new understanding of how early farmers adapted to a range of environmental and cultural settings.

One region that has received relatively little attention with respect to organic residue analysis is the Atlantic coast of Europe, with studies so far confined to Britain and Ireland[18,20]. Compared with other areas of Europe, this geographical unit has a particularly high density of Late Mesolithic sites[21], notably in Brittany, Denmark, along the Cantabrian coast of Spain and the Tagus and Sado estuaries of Portugal. Fish and shellfish were heavily exploited in these highly productive marine and estuarine ecotones during the Late Mesolithic period, immediately prior to the arrival of farming, e.g. refs. [22,23]. How and why farming became fully established along the Atlantic coastlines and estuaries replacing hunter–gatherer–fisher subsistence practices has been a source of much debate. One might expect that fish, shellfish or marine mammals were processed in the earliest Neolithic pottery reflecting continuity in economic practices particularly at sites located in areas close to where previous hunter–gatherer–fisher activity is recorded. However, paleodietary reconstructions of coastal Early Neolithic skeletal remains using stable isotope analysis appear to refute this hypothesis and show little evidence for the continued consumption of aquatic derived protein with the onset of the Neolithic[23,24]. Despite this, human remains from this period are extremely scarce in the coastal regions of the Iberian Peninsula and France, and this approach lacks the resolution to rule out marine foods entirely[25]. There is isotopic evidence from an Early Neolithic site in the Northern Isles of Britain for sporadic consumption of marine resources to supplement diets based largely on domesticated plants and animals[26]. In some cases, Mesolithic coastal communities co-existed with neighbouring farming villages[27] resulting in technological exchange[28], in other

areas Neolithic settlement clearly avoided Mesolithic territories although for much of the region the degree of farmer/forager interaction is debated[29]. The DNA evidence points to some limited admixture between foragers and farmers although precisely when and where this occurred is still difficult to discern[6,30].

A second question is the degree to which early agro-pastoral economies varied along the Atlantic European margin. The Neolithisation process of this region followed different rhythms and dynamics related to both maritime and continental influences, e.g. refs. [27–29,31,32]. In reality, the expansion of farming was relatively drawn out; pottery and domesticates appear some 1500 years later in the most northerly regions of the Atlantic Europe compared with the south. Thus, regional adaptations driven by local environmental factors and through interaction with different cultural groups may be expected in the intervening periods. Potential economic differences are partly indicated by the composition of faunal assemblages, which generally show a greater quantity of cattle remains relative to ovicaprines from the south to the north[19]. However, the scale of dairying by early farmers across Atlantic Europe compared with other regions of Europe is so far unknown.

To tackle these questions, here we provide new evidence from the organic residue analysis of 246 pottery sherds from 24 Early Neolithic sites situated between Portugal and Normandy as well as the Western Baltic (Supplementary Note 1). At each site, pottery representing only the initial phases of the Neolithic expansion were carefully chosen to create a representative corpus often from small and fragmented ceramic assemblages. We combine these data with previously published data from 39 archaeological sites[15,16,18,20,33,34] to create a supra-regional perspective on pottery use and animal exploitation by the earliest farmers (ca. 5500–3500 cal BC) (Fig. 1a, b). Together these data cover regions with variable densities of Mesolithic sites, and a latitudinal range that encompasses the entire Atlantic region from Portugal to Northern Scotland and the Western Baltic. Our results reveal an increased frequency of dairy products in Early Neolithic pottery assemblages correlating with their date and latitude.

## Results

**Molecular and isotopic results.** A total of 246 pottery sherds were analysed according to the well-established analytical procedures as described in 'Methods'. Interpretable amounts of lipids (i.e. >5 μg g$^{-1}$[35,36]) were obtained from 234 samples (95% of the total) (Supplementary Table 1). Excellent lipid preservation is in accordance with previous studies from the British Isles, Ireland and Northern Europe[15,18,20,37] and contrasts with studies undertaken in the Mediterranean and South-eastern Europe[9,10]. Molecular and isotopic data obtained from the acidified methanol and solvent extraction procedures are summarised in Supplementary Data 1 and 2.

The distribution of fatty acids recorded in pottery from the Iberian Peninsula and French sites, with relatively high amounts of $C_{18:0}$, cholesterol and the occasional presence of mid-chain ketones ($C_{33}$, $C_{35}$) and triacylglycerols ($C_{42}$–$C_{54}$), is consistent with degraded animal fats in the majority of cases (Supplementary Data 1). By measuring the stable carbon isotope ($\delta^{13}C$) values of the two main saturated fatty acids ($C_{16:0}$ and $C_{18:0}$), we were able to distinguish the source of these animal fats with greater certainty. This robust approach has been widely used for identifying fats derived from a wide range of sources and is based on physiological differences in fatty acid biosynthesis between tissues[37–39]. As well as absolute ranges (Fig. 2), ruminant adipose, ruminant dairy and porcine fats are distinguished according to differences in the carbon isotope values between the two main fatty acids ($\Delta^{13}C = \delta^{13}C_{18:0} - \delta^{13}C_{16:0}$; Fig. 2 in ref. [37]). In total,

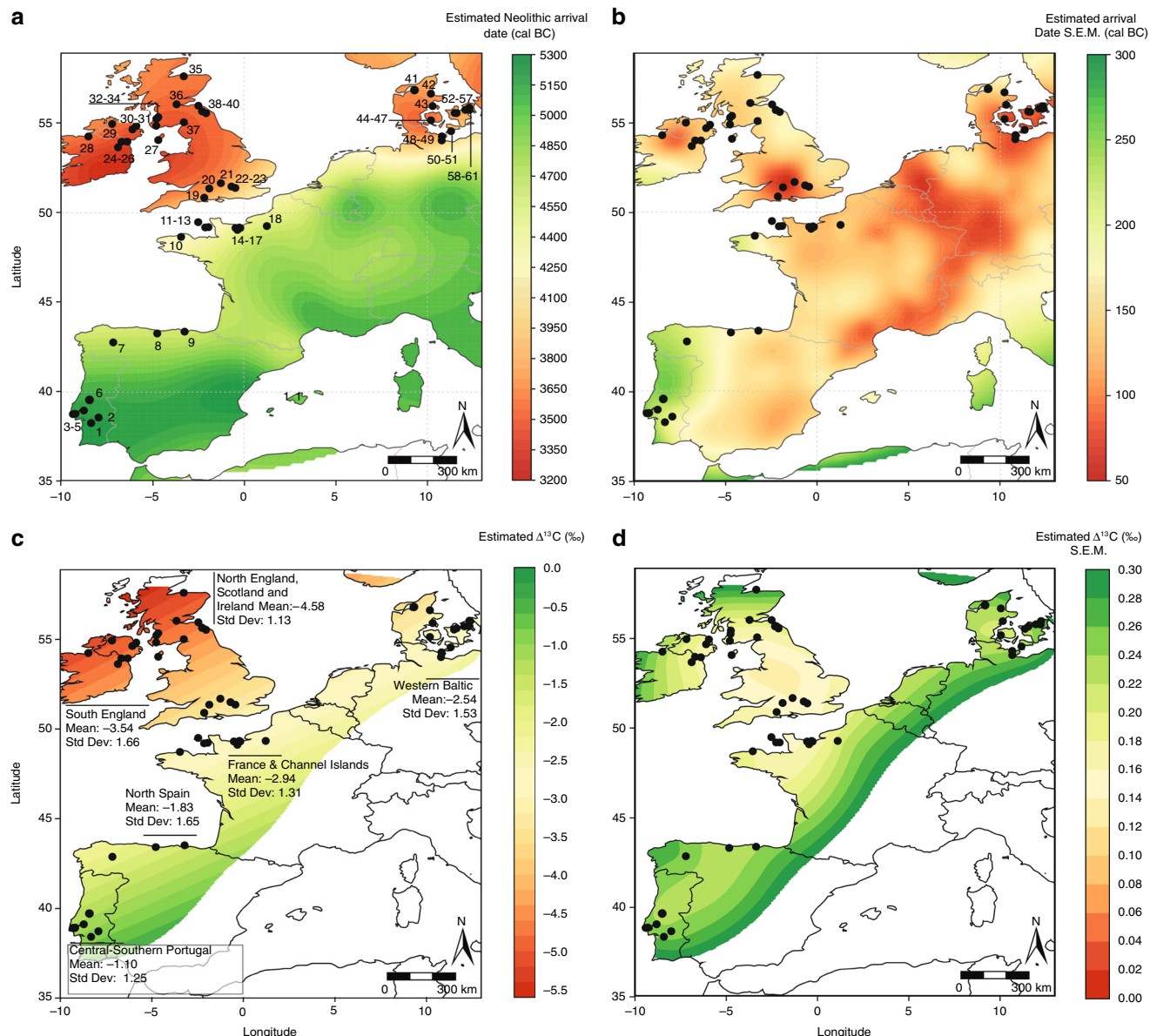

**Fig. 1 Map of Atlantic Europe showing estimated dates for the Neolithic dispersal and fatty acid carbon isotope compositions from Early Neolithic pottery. a** Output from SpreadR model showing estimated earliest arrival of the Early Neolithic and location of archaeological sites included in this paper (Supplementary Data 3). **b** Standard error of the mean for the arrival dates shown in **a**. **c** Output of AverageR model showing spatial estimates of $\Delta^{13}C$ values of fatty acids based on the analysis of 647 individual potsherds (Supplementary Data 2). Estimates are limited to areas where the standard error of the mean is <0.30‰. More negative $\Delta^{13}C$ values (i.e. ≤3.3‰) are typically associated with dairy fats, those between −3.3 and −1.0‰ are typical of ruminant fats and values ≥ −1‰ are typical of non-ruminant fats[38]. Mean and standard deviations of $\Delta^{13}C$ values by region are shown (Supplementary Data 3 provides these data for each site). **d** Standard error of the mean for the estimated $\Delta^{13}C$ values shown in **c**. Source data are provided as a Source Data file.

91.5% ($n = 225$) of the ceramic samples from Iberia, France and the Western Baltic provided sufficient lipids for analysis by GC-C-IRMS. These data were combined with existing datasets from Britain, Ireland and published data from the Western Baltic[15,16,18,20,33,34], creating a dataset of 647 samples (Supplementary Data 2), which were then compared with modern authentic reference samples of animal fats and oils (Supplementary Table 2, Fig. 2).

**Latitudinal gradient.** Interestingly, the proportion of dairy and ruminant carcass fats varies considerably between Early Neolithic assemblages across Atlantic Europe (Fig. 2a–f). There is a significant latitudinal gradient in the $\Delta^{13}C$ values (Spearman =

−0.67; $p = 7.8312E−75$) along the Western Atlantic coast from the Iberian Peninsula to Scotland with a greater proportion of potsherds from higher latitudes containing dairy products (Fig. 2c–e). Together, these data reveal different cultural and/or ecological scenarios for pottery use. Stable carbon isotope values typical of marine oils were only observed in Early Neolithic pottery from the Western Baltic (Fig. 2f) corresponding to the presence of aquatic lipid biomarkers in these vessels[16]. In this region, there is a degree of continuation in pottery use from the preceding hunter–gatherer–fishers of the ceramic Late Mesolithic Ertebølle culture[16,33]. In contrast, there was no evidence for aquatic biomarkers in any of the analysed vessels from sites located on the Atlantic seaboard (Supplementary Data 1), despite

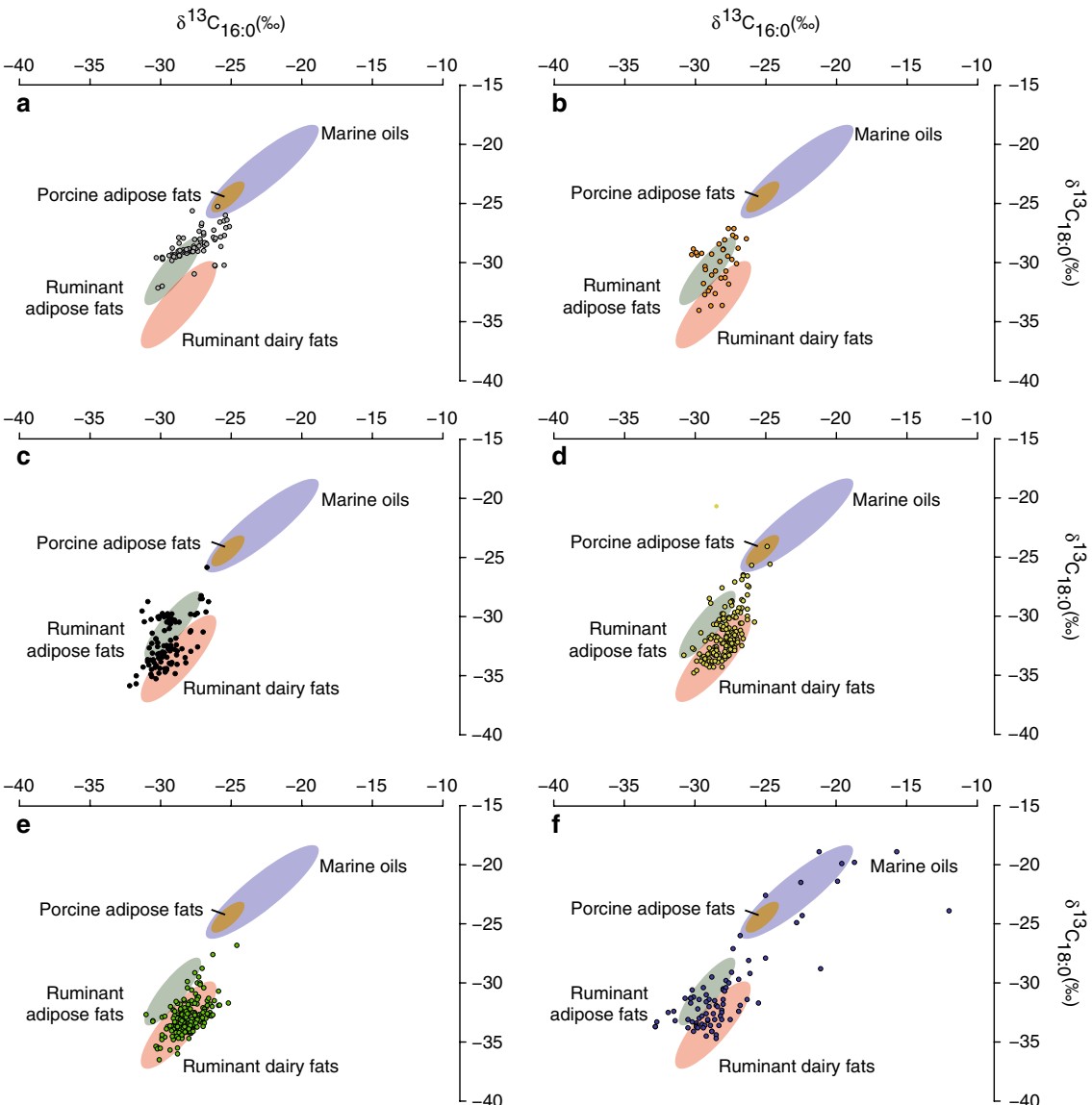

**Fig. 2 δ¹³C values of C₁₆:₀ and C₁₈:₀ *n*-alkanoic acids obtained from Early Neolithic pottery from the Atlantic coast of Europe and the Western Baltic (*n* = 647) (Supplementary Data 2). a** Central-southern Portugal. **b** Northern Spain. **c** France and the Channel Islands. **d** Southern England. **e** Northern England, Ireland and Scotland. **f** Western Baltic. The 68% confidence ellipses are based on modern European authentic reference fats and oils (ruminant adipose fats, ruminant dairy fats, porcine adipose fats and marine oils) (Supplementary Table 2). Note that several of the vessels from the Western Baltic fall outside of the ellipses and as such a mixture of resources, including marine oils with ruminant adipose/dairy fats is envisaged. Source data are provided as a Source Data file.

the use of highly sensitive protocols for their detection, and the fact that such compounds have been reported in much older samples from a wide range of environments, e.g. refs. [12,16,40]. Apart from one sample from the site of Lesmurdie Road in Scotland[18], no aquatic products were found in Early Neolithic pottery from Britain and Ireland using similar approaches[18,20]. This result either points to significantly reduced fishing and shellfish gathering at the start of the Neolithic or, given the occasional finds of fish remains and mollusc shell at some Neolithic sites, e.g. refs. [18,25,41,42] that aquatic resources were processed in other ways.

**Bayesian mixing model.** In order to highlight the potential effects of the mixing of different foodstuffs on the fatty acid stable isotope values, a Bayesian mixing model was deployed[43]. This approach seeks to examine any biases in the interpretation that

may arise due to variability in the fatty acid content and isotope values of potential contributing food sources. The model was applied to the median isotope values from each region (Supplementary Table 3), excluding the Western Baltic where the data are bimodally distributed between marine and terrestrial values. This approach highlights a high degree of equifinality in inferring pottery use based on the isotope data alone and it is often not possible to definitively exclude any of the potential source fats. Nevertheless, the model output (Fig. 3) shows that there is a clear increase in the proportion of lipids derived from dairy compared with meat northwards across the study transect area, confirming the pattern observed from consideration of the Δ¹³C values alone.

**Discussion**
Although we could not detect marine resources in any of the ceramic vessels with the exception of those from the Western

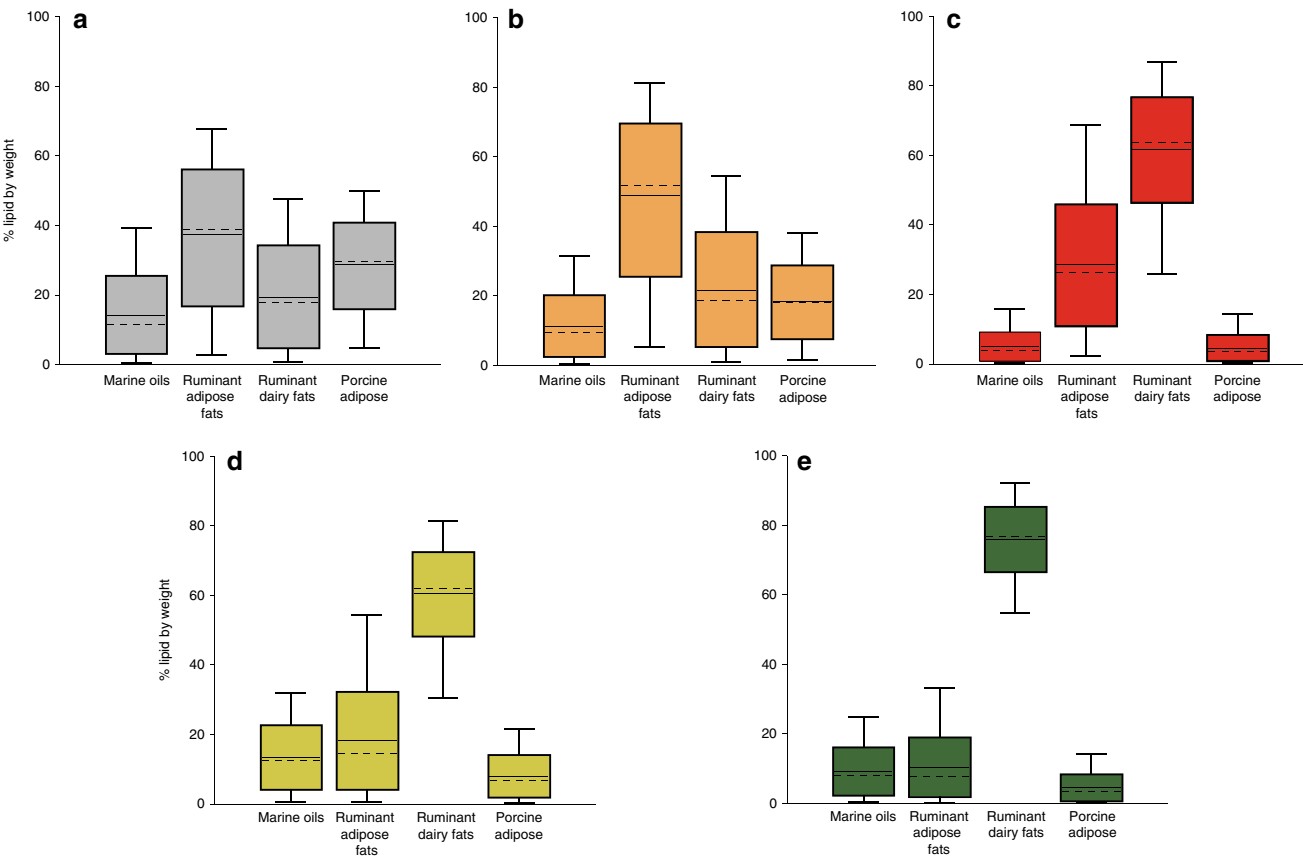

**Fig. 3 Bayesian modelled estimated proportions of lipids from different animal products in Early Neolithic pottery from different regions of Atlantic Europe (n = 563). a** Central-southern Portugal. **b** Northern Spain. **c** France and the Channel Islands. **d**. Southern England. **e** Northern England, Ireland and Scotland. The output of the model provides estimations of % lipid by weight (y-axis) based on the median $\delta^{13}C$ values of $C_{16:0}$ and $C_{18:0}$ $n$-alkanoic acids extracted from pots from each region with an uncertainty of 0.5‰ (Supplementary Table 3 and Supplementary Data 2). The boxes represent a 68% credible interval while the whiskers represent a 95% credible interval. The horizontal continuous line indicates the mean while the horizontal discontinuous line indicates the median. Source data are provided as a Source Data file.

Baltic, our results show considerable geographic differences in the use of ceramic artefacts along the Atlantic coast of Europe. The results also provide valuable insights into animal management strategies for their primary and secondary products, which is significant given the poor preservation of faunal remains, largely due to the prevailing acidic soils in these regions (Supplementary Table 4). Where available, Neolithic faunal data show that ovicaprines tend to dominate Iberian assemblages, whereas cattle were more important in Northern France, the UK and Denmark, e.g. refs. [8,19,41,44]. Based on the increased proportion of dairy residues associated with pottery from higher latitudes, we deduce that intensive dairying is closely linked with cattle-based economies, while sheep and goats were exploited for both their meat and milk, at least in the initial phases of the Neolithic. A similar association between cattle and dairying has been reported for the Early Neolithic of South-eastern Europe and the Near East[9] and may have been important for the initial expansion of farming beyond the Mediterranean climate zone[45].

Although we compare Early Neolithic assemblages across Atlantic Europe, defined by the first centuries following the appearance of pottery and domesticated animals, the sampling transect spans *ca.* 1500 years and follows the dispersal of farming, which appears *ca.* 5400/5300 cal BC in the south and *ca.* 3500 cal BC in the north (Fig. 1a, b). Therefore, the frequency of dairy residues in pottery is also correlated with the date that domesticates were introduced to any given area.

Along the Atlantic coast of Central and Southern Portugal, farming arrived with the spread of ovicaprine based economies of the Impressa–Cardial wares from the Mediterranean through the straits of Gibraltar[27,32] although other routes have been proposed for Northern Portugal and Galicia[46]. The analyses of Early Neolithic pottery assemblages from the Iberian sites point to economies oriented toward meat or mixed meat and milk production (Fig. 2a, b). Similarly, low frequencies of dairy lipids are observed in both Atlantic and Mediterranean Cardial pottery[10] supporting a mixed meat/milk economy during this early phase. Along the northern coast of Spain, farming and pottery were introduced through the Ebro valley or across the Western Pyrenees 500 years later, *ca.* 5000–4500 cal BC[29]. The increased frequency of dairy residues in this region may reflect a secondary Neolithisation front derived from Mediterranean populations (Epicardial) who had developed more intensified dairying practices.

In Atlantic France, pottery was obtained from sites north of the Loire river (Brittany and Normandy) dating to *ca.* 5300–4900 cal BC. Here, there is a clear technological and stylistic influence of final Rubané pottery that developed earlier in the Paris basin, which was linked with the cattle-based economies of the Central European Neolithic[47]. At these sites, pottery was used to process both ruminant milk and meat (Fig. 2c), which is consistent with the Early Neolithic cattle remains from Northern France and Central Europe that have mortality profiles indicative of a mixed meat/milk economy[48]. Ruminant carcass fats were also identified

in a small selection of 'La Hoguette' type potsherds from Alizay and Fontenay-le-Marmion, usually associated with indigenous foragers[49], although their function could not be discriminated from other Early Neolithic pots from the region.

In Western Britain and Ireland, a coastal Atlantic origin for the Neolithic (*ca.* 4000 cal BC) has long been argued for on cultural grounds[31], and is supported by new genetic evidence[6], although the precise origin and influence of the various Neolithic cultures of Northern France to the development of the British Neolithic is debated, and there is little evidence for direct cultural transmission[50]. Indeed, the spread of farming was 'delayed' for at least half a millennium following its arrival in adjacent regions on the European continental mainland. Over 80% of the Early Neolithic pottery from Britain and Ireland were found to contain dairy fats[18], twice that observed in North-western France and there is evidence for increased use of pottery for this purpose from Southern England to Scotland and Ireland (Figs. 1c, d and 2d, e). Its importance is further supported by recent findings of cattle milk proteins in the dental calculus of Early Neolithic individuals from the UK[51].

While the ecological limits on cereal production are often explicitly linked to the dispersal of farming[52], dairying also has specific requirements in terms of water availability, pasture quality and forage provision[53] and requires considerable expertise, especially in herd management. The reproductive timing of both cattle and sheep would have been further environmentally constrained[54], requiring cultural and biological adaptation to suit local conditions. Isotope analysis of cattle from Bercy in Northern France shows that the extension of their birthing season and their early weaning to suit more intensified dairying was established at least by the start of the 4th millennium cal BC[55,56], corresponding to the arrival of cattle into Britain. Milk and dairy products brought nutritional benefits, such as sources of fat and vitamin D. The latter may have been particularly critical for populations moving to higher latitudes where less of this vitamin is produced in vivo due to the reduced exposure to sunlight[57]. Dairying may also have been particularly important to farming populations struggling to establish cereal agriculture as they expanded into new territories sub-optimal for this purpose. Indeed, the available archaeobotanical evidence shows that the degree of arable farming in Britain varied considerably both locally and regionally during the Neolithic period[58], and in some regions it may even have failed following its initial introduction[59].

Along the Western Baltic coastlines of Northern Germany, Denmark and Southern Sweden, the first evidence for domesticated animals and plants appears at ca. 4000 cal BC associated with the emergence of the Early Neolithic Funnel Beaker culture (TRB), e.g. ref. [60]. At the same time there is a change in material culture, notably from Late Mesolithic Ertebølle to TRB pottery, but unlike other areas of Europe the exploitation of wild terrestrial game and fishing continued to be economically significant[16]. Nevertheless, even at these coastal sites, dairy products feature among the commodities present in these earliest TRB ceramics. Previous studies suggest that they were often processed separately, particularly in small beakers, flasks and bowls[61]. The variable use of pottery at coastal TRB sites encompassing both aquatic, dairy and other terrestrial resources may well be a consequence of the interaction of farmers and indigenous foragers. Further genomic analysis is needed to clarify the nature of such interactions but, here at least, indigenous and well-established culinary practices seemed to have persisted well into the Neolithic.

Overall, our study shows that when the 'Neolithic' arrived at different regions along the Atlantic coast there were different regional responses. We suggest these responses were influenced both by the different economic and cultural traditions of the

farmers who migrated to these new territories, the environments they moved into and the response of local foragers. Economic adaptations were needed before higher latitudes could be used for food production, resulting in hiatuses in the Neolithic expansion. Only in the Western Baltic, where a tradition of pottery use by Mesolithic hunter–gatherer–fishers was already established, were marine resources detected in Early Neolithic pottery. In the other locations, there is little evidence that the exploitation of coastal and estuarine environments had any influence on pottery use in the subsequent Early Neolithic. However, the frequency of dairy versus other terrestrial animal fats in pottery seems to be strongly influenced by latitude. Even in Britain and Ireland, where Early Neolithic sites in the south and the north have similar dates, dairy fats were more frequent at higher latitude sites perhaps highlighting the importance of local environmental conditions or nutritional requirements. Although Early Neolithic populations in Western Europe were largely lactose intolerant[62,63], variation in the scale of dairying observed across the Atlantic transect may have created a latitudinal gradient in selection pressure for adult lactase persistence (LP). A hypothesis supported by the high selection pressure for LP in North-western Europe inferred from its modern distribution[64], although subsequent large-scale episodes of migration, particularly in the Bronze Age[63], may also have influenced the LP distribution.

## Methods

**Sample information**. Pottery sherds were selected from archaeological sites from Portugal, Spain and France dated ca. 5500 and 3500 cal BC and the Western Baltic margin, dated between ca. 3950 and 3300 cal BC (Fig. 1a, b). AMS radiocarbon ($^{14}$C) dates attribute these sites to the Early Neolithic, i.e. contemporary with the earliest introduction of domesticated animals and plants. Descriptions of individual sites and samples are provided in Supplementary Note 1. Representative number of samples were taken related to the total number of sherds or ceramic vessels (Supplementary Table 1). In total, 246 samples were collected according to their morphological and decorative variability and their spatial distribution at each site.

**Lipid extraction and analytical protocol**. Lipids were extracted by direct transesterification from 246 pottery sherds using an acidified methanol protocol[35,65]. Briefly, methanol (4 mL) was added to 1 g of pottery powder and sonicated for 15 min. The suspension was acidified with sulfuric acid ($H_2SO_4$, 800 µL) and then heated for 4 h at 70 °C. Extraction of lipids was performed using *n*-hexane (3 × 4 mL). For quantification of lipids present in the resulting acid/methanol extracts (AEs), an internal standard (10 µL of hexatriacontane $C_{36:0}$) was added to all of the samples. Most of the samples from Iberia and France (185/224) were also extracted using DCM:MeOH (2:1, 3 × 2 mL) using established protocols, e.g. ref. [66]. The resulting total lipid extracts (TLEs) were dried under $N_2$ and derivatized with *N,O*-bis(trimethylsilyl)trifluoroacetamide (BSTFA) at 70 °C for 1 h. An additional internal standard (10 µL of hexatriacontane $C_{36:0}$) was added to all samples prior to analysis.

**Gas chromatography-flame ion detector**. GC-FID was carried out on AEs using an Agilent 7890S gas chromatograph (Agilent Technologies, Cheadle, Cheshire, UK). Samples were re-dissolved in hexane and 1 µL was injected into the GC at 300 °C with a splitless injector, using helium as carrier gas (2 mL min$^{-1}$). The GC column was a polymide coated fused-silica DB-1 (15 m × 320 µm × 0.1 µm; J&W Scientific, Folsom, CA, USA). The GC oven was set at 100 °C for 2 min, then increased by 20 °C min$^{-1}$ until 325 °C, where it was held for 3 min.

**Gas chromatography-mass spectrometry**. Analysis (AEs and TLEs) was carried out on an Agilent 7890A series chromatograph connected to an Agilent 5975 Inert XL mass detector (Agilent technologies, Cheadle, Cheshire, UK). Samples were injected with at splitless injector at 300 °C (1 µL) using helium as the carrier gas (constant flow, 3 mL min$^{-1}$). The spectra scanning window was between 50 and 800 *m/z* with a MS ionisation energy of 70 eV. A DB-5MS (50%-phenyl)-methylpolysiloxane column (30 m × 0.250 mm × 0.25 µm; J&W Scientific, Folsom, CA, USA) was used. The temperature program was 5 °C for 2 min, 10 °C min$^{-1}$ until 325 °C, followed by an isothermal hold for 15 min.

In order to identify ω-(*o*-alkylphenyl) alkanoic acids and isoprenoid fatty acids[67,68], and to calculate the ratio of phytanic acid diastereomers[69], analysis of the AEs (*n* = 241) was performed on a DB-23 (50%-cyanopropyl)-methylpolysiloxane column (60 m × 0.250 mm × 0.25 µm; J&W Scientific, Folsom, CA, USA). Briefly, samples were re-dissolved in hexane and 1 µL was injected with a splitless injector at 300 °C. Helium was used as the carrier gas with a constant flow of 3 mL min$^{-1}$.

The ionisation energy of the MS was 70 eV and spectra were obtained in SIM mode (74, 87, 88, 101, 105, 171, 213, 262, 290, 312, 318, 326, 346 $m/z$)[40]. The temperature profile was 50 °C for 2 min, 10 °C min$^{-1}$ until 100 °C, 4 °C min$^{-1}$ to 140 °C, 0.5 °C min$^{-1}$ to 160 °C, 20 °C min$^{-1}$ to 250 °C, with an isothermal hold for 10 min.

High temperature gas chromatography-mass spectrometry was performed on the TLEs ($n = 189$) using the same apparatus and conditions as above. Mass spectra were obtained by scanning between 50 and 1000 $m/z$ and analysis was performed with a HT-DB-1 100% dimethylpolysiloxane column (15 m × 0.320 mm × 0.1 μm; J&W Scientific, Folsom, CA, USA). The injector was maintained at 350 °C. The temperature program was 50 °C for 2 min, 10 °C min$^{-1}$ to 350 °C followed by an isothermal hold for 15 min.

**Gas chromatography-combustion isotope ratio mass spectrometry.** Compound specific isotope analysis was undertaken on the AEs of 225 samples. Stable carbon isotope values of methyl palmitate ($C_{16:0}$) and methyl stearate ($C_{18:0}$), derived from precursor fatty acids, were measured by GC-C-IRMS, following existing procedures[38]. The instrumentation consisted of an Agilent 7890B series GC (Agilent Technologies, Santa Clara, CA, USA) linked by an Isoprime GC5 interface (Isoprime Cheadle, UK) to an Isoprime 100 (Isoprime, Cheadle, UK) and to an Agilent 5975C inert mass spectrometer detector (MSD). Samples were re-dissolved and 1 μL was injected into DB-5MS ultra-inert fused-silica column (60 m × 0.250 mm × 0.25 μm, J&W Scientific, Folsom, CA, USA). The temperature program was 50 °C for 0.5 min, 25 °C min$^{-1}$ to 175 °C, 8 °C min$^{-1}$ to 325 °C, isothermal hold for 20 min. The carrier gas used was ultra-high purity grade helium (3 mL min$^{-1}$). The gas flow eluting from the column was split into two streams. One was directed to the MSD for compound identification, while the other was directed through the CuO furnace tube at 850 °C to convert all the carbon species to $CO_2$. Ion intensities (44, 45 and 46 $m/z$) of eluted products were recorded and the corresponding $^{13}C/^{12}C$ ratios were computed.

Data analysis (IonVantage and IonOS software; Isoprime, Cheadle, UK) made comparisons between samples and a standard reference gas ($CO_2$) of known isotopic composition. The results are expressed in per mill (‰) relative to an international standard, VPDB. Within each batch, a mixture of $n$-alkanoic acid ester standards of known isotopic composition (Indiana standard F8-3) was used to check instrument accuracy (<0.3‰) and precision (<0.5‰). Each sample was measured at least in duplicate (mean S.D. = 0.1‰ for $C_{16:0}$ and 0.1‰ for $C_{18:0}$). The resulting data were corrected to account for methylation through comparisons with $C_{16:0}$ and $C_{18:0}$ fatty acid standard of known isotopic composition that were processed with each batch under identical conditions. $\delta^{13}C$ values obtained from a range of European authentic adipose fats, dairy fats and marine oils were collated and corrected for the Suess effect[70] taking into consideration the date of collection (Supplementary Table 2).

**Mixing model and spatial interpolation.** Modelling was carried out using the 3.0 Beta version of the Bayesian mixing model FRUITS[71] (available at http://sourceforge.net/projects/fruits/). The model was implemented using $\delta^{13}C_{16:0}$ and $\delta^{13}C_{18:0}$ as proxies. Four food groups were selected as potential sources (marine, ruminant adipose fats, ruminant dairy fats and non-ruminant fats) and $\delta^{13}C$ values for each were obtained from modern authentic reference fats and oils (Supplementary Table 2). Uncertainties were derived using a covariance matrix and standard errors of the mean $\delta^{13}C$ values for each food source, assuming that the vessels were used repeatedly. Palmitic and stearic acid concentration values (Supplementary Table 2) were obtained from the USDA Food Composition Databases (https://ndb.nal.usda.gov/ndb/). Uncertainties were derived from the standard error of the mean values. The concentrations and model outputs are expressed as % of total lipid by weight. Numerical Bayesian inference was performed using the BUGS software, a Markov chain Monte Carlo (MCMC) method that employs Gibbs sampling and the Metropolis–Hastings algorithm. The first 5000 iterations of the MCMC chains were discarded (burn-in steps) and these were then run for an additional 10,000 iterations. Model convergence was checked by inspecting if the trace plots of the respective posterior chains exhibited an asymptotic behaviour. Trace autocorrelation plots were also inspected to assess convergence.

To generate spatial estimates of $\Delta^{13}C$ values and to estimate the earliest arrival of the Neolithic we employed the AverageR and SpreadR models, respectively. These are available as R-based[72] Open Access apps (https://www.isomemoapp.com/) developed within the Pandora & IsoMemo initiatives.

AverageR is a generalized additive mixed model that uses a thin plate regression spline[73]. This spline smoother employs a Bayesian smoothing parameter governing the smoothness of the surface, which is estimated from the data and trades-off bias against variance to make the optimal prediction for new, unseen data[73–75]. By introducing a random intercept for the site, intra-site as well as the inter-site variation were employed in estimating uncertainty, expressed as a standard error of the mean. More specifically, we employ the following modelling formula:

$$Y_{ij} = s(longitude, latitude) + u_i + \varepsilon_{ij},$$

where

$Y_{ij}$: independent variable for site I and individual j.
$s(longitude, latitude)$: spline smoother[73].
$u_i \sim N(0, \sigma_u)$: random intercept for site i.

$\varepsilon_{ij} \sim N(0, \sigma_\varepsilon)$: residual error for individual j in site i.
SpreadR employs an extreme quantile approach by replacing the minima or first arrival with a 0.5%—quantile[76]. This quantile was estimated using Bayesian quantile regression using a spline smoother[77]. Estimated uncertainty, expressed as standard error of the mean, should be viewed as a low uncertainty estimate which may underestimate the true uncertainty[77].

**Reporting summary.** Further information on research design is available in the Nature Research Reporting Summary linked to this article.

## Data availability

The authors declare that all data supporting this research are available within the paper, its Supplementary Information and Supplementary Data files. Source data underlying Figs. 1, 2 and 3 are provided as a Source Data file. EUROVOL dataset has been used in the creation of Fig. 1 a, b (https://discovery.ucl.ac.uk/id/eprint/1469811/) and USDA Food Composition Databases (https://ndb.nal.usda.gov/ndb/) have been used to obtain concentration of $C_{16:0}$ and $C_{18:0}$ $n$-alkanoic acids in the different products (Supplementary Table 2).

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

## Acknowledgements

We thank Carl Heron for permission to include the stable carbon isotope values from the two Rødbyhavn sites. This work was supported by the European

Commission [CerAM -653354- H2020-MSCA-IF-2014]; UK Arts and Humanities Research Council Grant [AH/E008232/1] and the British Academy [R1850601].

## Author contributions

M.C. and O.E.C. designed the research; M.C., A.L., H.K.R. and A.C.C. performed the research; M.C., A.L., H.K.R., R.F. and O.E.C. analysed the data; M.C., A.L., H.K.R., A.C.C., R.F. and O.E.C. wrote the paper. P.A., B.A., C.B., D.J., M.D., R.F.V., C.G.V., L.J., A.L.H., C.M., S.M., G.M., C.N, R.O.P., X.P.R.A., T.S. and J.Z. provided samples and contextual information. All the authors were involved in reviewing the manuscript.

## Competing interests

The authors declare no competing interests.
