## [Peer Review File · Nature Communications]

Reviewers' Comments:

Reviewer #1:

Remarks to the Author:

This manuscript presents data from the analyses of lipid residues preserved in pottery vessels from the Atlantic coast. It compiles new analyses (n = 246) together with work previously published by the same group and other groups (> 400 sherds). This manuscript suggests that the authors are in a position provide a map of $\Delta^{13}\text{C}$ values across Europe, relating to identification of milk use, however, the data fall some way short of being able to achieve this.

The data is treated statistically using FRUITS to assess intervals of confidence in the proportion of various animal products. I do not use this software so am unable to comment on this part of the manuscript.

I have a number of general concerns about this manuscript:

1. The authors present their work as new, emphasising that their results are "surprising" (l. 65) and that the region of interest in this study has only received "little attention" (l. 107). However, the absence of aquatic foods has already been reported by other authors using part of the dataset presented in this manuscript. Indeed > 400 sherds presented in this study have already been published, and similar conclusions had been reached in recent publications.
2. The authors present the results of analyses 600+ sherds (including 400+ already published data) for the whole of the northern European Atlantic coast, which has a distance of many thousands of km. The sampling rate is therefore extremely low. The shortcomings of the sampling are emphasised by a range of other recent regional studies involving lipid residue analyses published in the recent years. Most studies have looked at 1,000+ sherds, often over smaller geographical regions.
3. Also, the authors propose that intensive dairying is linked with cattle-based agriculture, citing some archaeozoological studies. However, the authors do not present any archaeozoological data in the manuscript (l. 250-252). Critically, other recent regional studies have correlated the lipid residue findings with detailed archaeozoological assessments.
4. Importantly, although the authors are focussing on the Atlantic coast, the map they are presenting as Figure 1 is showing interpolated data for the entire Western Europe. This is extremely misleading and should be amended. The authors should be interpolating the data only along the Atlantic coast as their data cover this geographical area.

In brief, the manuscript is relying on a too low number of sherds to make inferences on animal management at such a broad regional level. Further, the authors do not back up their conclusions with detailed archaeozoological assessments. Finally, the manuscript strays into a number of other conclusions which are unsupported by data, e.g. lactase persistence.

The field of organic residues is moving fast, and such a paper would seem to be inappropriate for publication in such a high-impact journal such as Nature Communications. The manuscript would be more suited to a specialised journal such as Journal of Archaeological Science (JAS) or JAS Reports, although that said there is no covering up the deficiencies in the research design.

Reviewer #2:

Remarks to the Author:

This paper presents an impressive dataset from analysis of lipids in pottery, and successfully combines new and already published data, to come up with a very broad geographical look at the early Neolithic in Atlantic Europe, and the development of cattle keeping and dairying. Its headlines are the increase in both cattle keeping and dairying with increasing latitude, along with a near-absence of signs of use of aquatic resources as detectable in lipid analysis of pottery. Overall, I would predict considerable impact and a wide readership, even if some of these headlines (cattle dominance as one goes north, and reduced or minimal aquatic resource use) have already been indicated from conventional faunal studies and other scientific analyses. The relevance of the study to adult lactase persistence is another point well made. The paper is well referenced and illustrated, and the supplementary information seems both detailed and rigorous.

I would certainly strongly recommend its publication, though I think the manuscript can be improved.

Your geographical transect is impressively broad, but it might give the uninitiated the impression that the spread of Neolithic people and practices took place on a simple south-north gradient. You are obviously aware of the early Neolithic in central Europe from the middle of the sixth millennium cal BC onwards but I suggest that you show that more clearly in the initial part of the paper (and could cite recent and ongoing research on milk/dairy use in the LBK). Central Europe comes in again later (eg line 280f) and again I felt that a little more explanation of the chronology and the relation with the West Baltic/south Scandinavian region would have helped to make things clearer.

At times in the discussion (eg lines 260f, 318f) you veer into rather fuzzy generalisation, whereas the detailed region-by-region discussions are much more effective. On that note, however, the treatment of the West Baltic in your discussion seems rather circumscribed, and could be expanded a little, space permitting.

There is an inherent limitation in the genre of papers for Nature and some other leading scientific journals, in that there is space for condensed presentation of large datasets and their analysis, but less room for discussion of complex issues of interpretation and implications. Your paper illustrates this rather well. You refer in the closing discussion to a possible range of explanatory factors (eg line 320 'different economic and cultural traditions') but this could involve all manner of things, and one could wish for a much more detailed climate record against which to compare your data, as well as, in an ideal/non-space restricted world, more detail on individual archaeological contexts as opposed to the broad regionalised pictures given. The way through this if you are redrafting parts of the paper may simply be to acknowledge the complexity of the many issues raised, and to signal the need for ongoing detailed discussion elsewhere.

Lastly, a few details.

Line 315: I would rephrase 'cereal agriculture failed' as there are complex issues surrounding this question and the trajectory may not be the same everywhere across Britain and Ireland.

Typos/syntax:

Line 121 needs better punctuation

Line 262 'transects' = transect

Line 288 'forager' = foragers

Line 335 ?should/could read 'can be inferred'

Line 337 'influence' = influenced

Reviewer #3:

Remarks to the Author:

Overall this is an excellent paper presenting important new data on lipid evidence for dairying, noting a latitudinal gradient. Given that modern lactase persistence shows a comparable trend, this is perhaps not surprising, but it is very useful to see it demonstrated so convincingly for the early Neolithic in such a wide-scale study.

A minor quibble is that the section setting out of the research context underplays the body of research on the transition along the Atlantic façade, including those addressing the importance of dairying from a faunal perspective (see especially the work of Anne Tresset). Human DNA is mentioned on a number of occasions, but there is also a considerable body of cattle DNA for the region that is relevant for setting the background scene.

107/ the statement that the Meso-Neolithic transition on the Atlantic coast has received relatively little attention (isotopically?) downplays a considerable body of stable isotopic, faunal and material culture research on this question (Arias, Dupont, Kador, Marchand, Montgomery, Richards, Scarre, Schulting, Sheridan, Smits, Tresset, Woodman, etc.).

122-4/ the evidence for isotopic evidence from the 'Northern Isles' of Britain for the continued exploitation of marine foods is questionable. Two references are cited in support. The Montgomery et al. paper refers to only sporadic (this word even features in the paper's title) use probably during periods of crop failure in a very marginal environment for farming, while the Charlton et al. paper refers to a culturally entirely Mesolithic site (albeit a very late one, probably overlapping with Neolithic sites elsewhere in Britain) with no evidence of domesticated plants or animals or of Neolithic material culture (e.g., pottery). Also the site is not in the Northern Isles.

197/ the probable geographical/environmental difference underlying the results for Portugal and Spain in Fig. 2 might be emphasised by specifying these as 'central/southern Portugal' and 'northern Spain' since from Figure 1 this is where the samples from these countries originated.

Fig. 2F, can comment on why a small number of samples fall so far outside of the modern reference ellipse for marine oils, in directions that do not seem to follow any mixing line between this and the other reference materials?

237/ Why does the Bayesian modelling in Fig. 3 not include the Baltic area?

310/ ungrammatical sentence beginning 'The latter...']

315/ Might acknowledge that there is still debate concerning the putative failure of arable farming in EN Britain based on the scarcity of cereals in archaeobotanical assemblages (cf. Bishop 2015).

Dear editor,

We are grateful for considering our manuscript entitled "*Latitudinal gradient in dairy production with the introduction of farming in Atlantic Europe*" for publication in *Nature Communications*. Our manuscript comprises a main text, three figures and Supplementary Information. In our reviewed version, we have included some modifications related to the comments and suggestions made by the three reviewers. In addition, we would like to include some additional comments to the general concerns addressed by the reviewers. First, we include the complete comments of the reviewers. Subsequently, we include our point by point to these comments.

Reviewers' comments:

Reviewer #1 (Remarks to the Author):

This manuscript presents data from the analyses of lipid residues preserved in pottery vessels from the Atlantic coast. It compiles new analyses (n = 246) together with work previously published by the same group and other groups (> 400 sherds). This manuscript suggests that the authors are in a position provide a map of $\Delta^{13}\text{C}$ values across Europe, relating to identification of milk use, however, the data fall some way short of being able to achieve this.

The data is treated statistically using FRUITS to assess intervals of confidence in the proportion of various animal products. I do not use this software so am unable to comment on this part of the manuscript.

I have a number of general concerns about this manuscript:

1. The authors present their work as new, emphasising that their results are “surprising” (l. 65) and that the region of interest in this study has only received “little attention” (l. 107). However, the absence of aquatic foods has already been reported by other authors using part of the dataset presented in this manuscript. Indeed > 400 sherds presented in this study have already been published, and similar conclusions had been reached in recent publications.

2. The authors present the results of analyses 600+ sherds (including 400+ already published data) for the whole of the northern European Atlantic coast, which has a distance of many thousands of km. The sampling rate is therefore extremely low. The shortcomings of the sampling are emphasised by a range of other recent regional studies involving lipid residue analyses published in the recent years. Most studies have looked at 1,000+ sherds, often over smaller geographical regions.

3. Also, the authors propose that intensive dairying is linked with cattle-based agriculture, citing some archaeozoological studies. However, the authors do not present any

archaeozoological data in the manuscript (l. 250-252). Critically, other recent regional studies have correlated the lipid residue findings with detailed archaeozoological assessments.

4. Importantly, although the authors are focussing on the Atlantic coast, the map they are presenting as Figure 1 is showing interpolated data for the entire Western Europe. This is extremely misleading and should be amended. The authors should be interpolating the data only along the Atlantic coast as their data cover this geographical area.

In brief, the manuscript is relying on a too low number of sherds to make inferences on animal management at such a broad regional level. Further, the authors do not back up their conclusions with detailed archaeozoological assessments. Finally, the manuscript strays into a number of other conclusions which are unsupported by data, e.g. lactase persistence.

The field of organic residues is moving fast, and such a paper would seem to be inappropriate for publication in such a high-impact journal such as Nature Communications. The manuscript would be more suited to a specialised journal such as Journal of Archaeological Science (JAS) or JAS Reports, although that said there is no covering up the deficiencies in the research design.

Reviewer #2 (Remarks to the Author):

This paper presents an impressive dataset from analysis of lipids in pottery, and successfully combines new and already published data, to come up with a very broad geographical look at the early Neolithic in Atlantic Europe, and the development of cattle keeping and dairying. Its headlines are the increase in both cattle keeping and dairying with increasing latitude, along with a near-absence of signs of use of aquatic resources as detectable in lipid analysis of pottery. Overall, I would predict considerable impact and a wide readership, even if some of these headlines (cattle dominance as one goes north, and reduced or minimal aquatic resource use) have already been indicated from conventional faunal studies and other scientific analyses. The relevance of the study to adult lactase persistence is another point well made. The paper is well referenced and illustrated, and the supplementary information seems both detailed and rigorous.

I would certainly strongly recommend its publication, though I think the manuscript can be improved.

Your geographical transect is impressively broad, but it might give the uninitiated the impression that the spread of Neolithic people and practices took place on a simple south-north gradient. You are obviously aware of the early Neolithic in central Europe from the middle of the sixth millennium cal BC onwards but I suggest that you show that more clearly in the initial part of the paper (and could cite recent and ongoing research on milk/dairy use in the LBK). Central Europe comes in again later (eg line 280f) and again

I felt that a little more explanation of the chronology and the relation with the West Baltic/south Scandinavian region would have helped to make things clearer.

At times in the discussion (eg lines 260f, 318f) you veer into rather fuzzy generalisation, whereas the detailed region-by-region discussions are much more effective. On that note, however, the treatment of the West Baltic in your discussion seems rather circumscribed, and could be expanded a little, space permitting.

There is an inherent limitation in the genre of papers for Nature and some other leading scientific journals, in that there is space for condensed presentation of large datasets and their analysis, but less room for discussion of complex issues of interpretation and implications. Your paper illustrates this rather well. You refer in the closing discussion to a possible range of explanatory factors (eg line 320 'different economic and cultural traditions') but this could involve all manner of things, and one could wish for a much more detailed climate record against which to compare your data, as well as, in an ideal/non-space restricted world, more detail on individual archaeological contexts as opposed to the broad regionalised pictures given. The way through this if you are redrafting parts of the paper may simply be to acknowledge the complexity of the many issues raised, and to signal the need for ongoing detailed discussion elsewhere.

Lastly, a few details.

Line 315: I would rephrase 'cereal agriculture failed' as there are complex issues surrounding this question and the trajectory may not be the same everywhere across Britain and Ireland.

Typos/syntax:

Line 121 needs better punctuation

Line 262 'transects' = transect

Line 288 'forager' = foragers

Line 335 ?should/could read 'can be inferred'

Line 337 'influence' = influenced

Reviewer #3 (Remarks to the Author):

Overall this is an excellent paper presenting important new data on lipid evidence for dairying, noting a latitudinal gradient. Given that modern lactase persistence shows a comparable trend, this is perhaps not surprising, but it is very useful to see it demonstrated so convincingly for the early Neolithic in such a wide-scale study.

A minor quibble is that the section setting out of the research context underplays the body of research on the transition along the Atlantic façade, including those addressing the importance of dairying from a faunal perspective (see especially the work of Anne Tresset). Human DNA is mentioned on a number of occasions, but there is also a considerable body of cattle DNA for the region that is relevant for setting the background

scene.

107/ the statement that the Meso-Neolithic transition on the Atlantic coast has received relatively little attention (isotopically?) downplays a considerable body of stable isotopic, faunal and material culture research on this question (Arias, Dupont, Kador, Marchand, Montgomery, Richards, Scarre, Schulting, Sheridan, Smits, Tresset, Woodman, etc.).
122-4/ the evidence for isotopic evidence from the ‘Northern Isles’ of Britain for the continued exploitation of marine foods is questionable. Two references are cited in support. The Montgomery et al. paper refers to only sporadic (this word even features in the paper’s title) use probably during periods of crop failure in a very marginal environment for farming, while the Charlton et al. paper refers to a culturally entirely Mesolithic site (albeit a very late one, probably overlapping with Neolithic sites elsewhere in Britain) with no evidence of domesticated plants or animals or of Neolithic material culture (e.g., pottery). Also the site is not in the Northern Isles.

197/ the probable geographical/environmental difference underlying the results for Portugal and Spain in Fig. 2 might be emphasised by specifying these as ‘central/southern Portugal’ and ‘northern Spain’ since from Figure 1 this is where the samples from these countries originated.

Fig. 2F, can comment on why a small number of samples fall so far outside of the modern reference ellipse for marine oils, in directions that do not seem to follow any mixing line between this and the other reference materials?

237/ Why does the Bayesian modelling in Fig. 3 not include the Baltic area?
310/ ungrammatical sentence beginning ‘The latter...’]

315/ Might acknowledge that there is still debate concerning the putative failure of arable farming in EN Britain based on the scarcity of cereals in archaeobotanical assemblages (cf. Bishop 2015).

Response to reviewer 1:

1. The authors present their work as new, emphasising that their results are “surprising” (l. 65) and that the region of interest in this study has only received “little attention” (l. 107). However, the absence of aquatic foods has already been reported by other authors using part of the dataset presented in this manuscript. Indeed > 400 sherds presented in this study have already been published, and similar conclusions had been reached in recent publications.

To our knowledge we present the first data regarding pottery use from the Atlantic coast of France, Spain and Portugal. As explained in the manuscript, the results are surprising as the expectation was for some continuity from the Mesolithic given its extent and importance, particularly in southern-central Portugal and northern Spain. Also we emphasise the importance of a latitudinal gradient in dairying which has not been hitherto reported in previous studies of British or Irish Neolithic pottery.

2. The authors present the results of analyses 600+ sherds (including 400+ already published data) for the whole of the northern European Atlantic coast, which has a distance of many thousands of km. The sampling rate is therefore extremely low. The shortcomings of the sampling are emphasised by a range of other recent regional studies involving lipid residue analyses published in the recent years. Most studies have looked at 1,000+ sherds, often over smaller geographical regions.

We thank the reviewer for this observation and we have had the opportunity to respond to this exact point in a previous review of this article. To assess the sampling rate presumes knowledge of the extent of material available. In fact, the amount of pottery specifically dating to the 'Early Neolithic' is low compared to later prehistoric periods and '1,000+ sherds' are simply not available, in part due to the character of the archaeological record. Moreover, for some of the selected sites we sampled all of the available pottery. We had estimated the sampling rate at each site given the extent of the assemblage, where this is known (see Supplementary Table 1) and one can see that it is reasonably high.

We also include the following sentence in the text "*At each site, pottery representing only the initial phases of the Neolithic expansion were carefully chosen to create a representative corpus often from small and fragmented ceramic assemblages.*" (Lines 139-141)

3. Also, the authors propose that intensive dairying is linked with cattle-based agriculture, citing some archaeozoological studies. However, the authors do not present any archaeozoological data in the manuscript (l. 250-252). Critically, other recent regional studies have correlated the lipid residue findings with detailed archaeozoological assessments.

We thank the reviewer for this observation and we have had the opportunity to respond to this exact point in a previous review of this article. If the reviewer refers to the Supplementary Table 3, they will see that we had included information about the number of faunal remains, including NISP values, when available for each site.

Unfortunately, we were unable to undertake the correlations the referee desires due the lack of data resulting from the poor preservation of faunal remains from Neolithic sites along the Atlantic coast.

Indeed, this topic has been recently addressed for Portugal due to the poor preservation and/or lack of published reports (Valente and Carvalho 2014). Similarly, faunal data for many of the French sites, published in sites monographies or scientific reports

(Ghesquiére and Aubry 2013; Billard et al. 2014; Juhel 2015; Germain-Vallée 2015), also demonstrates the poor preservation of organic remains that prevails in this region.

To clarify this point, in the discussion we noted that *‘The results also provide valuable insights into animal management strategies for their primary and secondary products, which is significant given the poor preservation of faunal remains, largely due to the prevailing acidic soils in these regions (Supplementary Table 3).’* (Lines 240-243)

References

Billard, C., Bostyn, F., Hamon, C. & Meunier, K. L’habitat du Néolithique ancien de Colombelles ‘Le Lazzaro’ (Calvados). Mémoires de la Société Préhistorique Française (Société Préhistorique Française, 2014).

Germain-Vallee, C. Un hameau du Néolithique ancien et les indices d’une occupation du Néolithique Moyen II. (2015, Scientific Report DGA- Education, Jeunesse et Culture. Service Archéologie).

Ghesquiére, E. & Aubry, B. in *Transitions, ruptures et continuité en Préhistoire* 503-522 (Société Préhistorique Française 2013).

Juhel, L. Un habitat du Néolithique ancien. Rapport final d’opération. (2015).

Valente, M. J. Zooarqueologia do Neolítico do Sul de Portugal: passado, presente e futuros. in *O Neolítico em Portugal antes do Horizonte 2020: perspectivas em debate* (eds. Diniz, M., Neves, C. & Martins, A.) 87–108 (Associação dos Arqueólogos Portugueses, 2015).

4. Importantly, although the authors are focussing on the Atlantic coast, the map they are presenting as Figure 1 is showing interpolated data for the entire Western Europe. This is extremely misleading and should be amended. The authors should be interpolating the data only along the Atlantic coast as their data cover this geographical area.

This is an important point raised by the reviewer. We argue that the map is not-misleading as we also included the uncertainty of the interpolation for the whole region we consider. Providing the accompanying uncertainty map provides transparency and robustness which we agree is rare when data of these nature are presented. However, we are keen to avoid any misinterpretations of our dataset so we have redrawn the map to address the concerns raised by the reviewer #1 and now only report data where the uncertainty on the $\Delta^{13}\text{C}$ measurements are $< 0.3 \text{ ‰}$. Also, we have modified the caption for Figure 1:

Fig. 1. Map of Atlantic Europe showing radiocarbon (^{14}C) dates for the Neolithic dispersal and the fatty acid carbon isotope compositions from Early

Neolithic pottery. A. Approximate radiocarbon dates for the arrival of the Early Neolithic (Supplementary Source Data 1) and location of archaeological sites included in this paper. **B.** Uncertainties of the radiocarbon dates (range of calibrated dates). **C.** Spatial interpolation of the distribution of $\Delta^{13}\text{C}$ values of fatty acids from 647 individual potsherds. The interpolation is limited to areas with a standard error lower than 0.30‰. More negative $\Delta^{13}\text{C}$ values (i.e. $< -3.3\text{‰}$) are typically associated with dairy fats, those between -3.3‰ and -1.0‰ are typical of ruminant fats and values $> -1\text{‰}$ are typical of non-ruminant fats⁴⁰. **D.** $\Delta^{13}\text{C}$ interpolation standard error.

Related to this figure, we have added in Supplementary Information one section “Supplementary Source data 1”. On it we compiled the bibliographical references to elaborate the Figure 1A and B and the criteria for calibration.

5. Finally, the manuscript strays into a number of other conclusions which are unsupported by data, e.g. lactase persistence.

The production of dairy products is a prerequisite for the selection of adult lactase persistence. Therefore, geographic differences in dairy production are clearly relevant to discussion regarding the selection of this trait.

Response to reviewer 2:

1. Your geographical transect is impressively broad, but it might give the uninitiated the impression that the spread of Neolithic people and practices took place on a simple south-north gradient. You are obviously aware of the early Neolithic in central Europe from the middle of the sixth millennium cal BC onwards but I suggest that you show that more clearly in the initial part of the paper (and could cite recent and ongoing research on milk/dairy use in the LBK). Central Europe comes in again later (eg line 280f) and again I felt that a little more explanation of the chronology and the relation with the West Baltic/south Scandinavian region would have helped to make things clearer.

The map (Figs. 1A and B) plots the locations of the sampled sites against the dates for the dispersal of the Neolithic (domesticated animals and plants) in Europe, which is clearly not on a S-N trajectory. Concerning the chronological differences, they are addressed in the main text but we have amended the sentence to make sure we are referring specifically to the Atlantic coast rather than the rest of Europe, “*In reality, the expansion of farming was relatively drawn out; pottery and domesticates appear some 1,500 years later in the most northerly regions of Atlantic Europe compared to these to the south*” (lines 127-130) and Figs. 1A and B.

Radiocarbon dates are older in south-central Portugal where the introduction of the Neolithic is related to Impressa-Cardial wares from the Mediterranean (lines 260-263, Fig 1A). The earliest archaeological sites, such as Lapa Lameiras and Gruta do Caldeirão, are included in this article (see additional archaeological information). Available dates for northern Spain have a later chronology, *ca.* 5000-4500 cal BC (Fig. 1A), although in this case two possible routes of influence on the Neolithisation process have been processed: (i) influence of the Cardial ware though the Ebro valley; and (ii) influence across the western Pyrenees (lines 267-269). The chronology for the introduction of farming and livestock in Atlantic France is not homogenous. The earliest Neolithic archaeological sites are well dated to *ca.* 5300-4900 cal BC (lines 273-274), and include Colombelles “Le Lazzaro”, Verson “Les Mesnils” and Lannion “Kervouric”, which are included in this study (additional archaeological information).

Several projects have been carried out based on LBK archaeozoological remains (Gillis et al. 2017 -already included in the references-) and organic residue analysis (e.g. Salque et al. 2012; Salque et al. 2013). Moreover, we have included further references in the 2nd paragraph to refer more specifically to dairying in Central Europe as suggested by the referee:

Although milk derived lipids have now been identified in Early Neolithic pottery from southern^{9,10}, central¹¹⁻¹³ and northern Europe¹⁵⁻¹⁸, the scale and intensity of dairying in relation to meat production is still unknown. (Lines 93-95)

Finally, we address the issue raised concerning the Western Baltic below.

References

Gillis, R. E. *et al.* The evolution of dual meat and milk cattle husbandry in Linearbandkeramik societies. *Proceedings of the Royal Society B* **284**, 20170905, doi:10.1098/rspb.2017.0905 (2017).

Salque, M. *et al.* Earliest evidence for cheese making in the sixth millennium BC in northern Europe. *Nature* **493**, 522–525, doi: 10.1038/nature11698 (2013).

Salque, M. *et al.* New insights into the Early Neolithic economy and management of animals in Southern and Central Europe revealed using lipid residue analyses of pottery vessels. *Anthropozoologica* **47**, 45–62, doi. 10.5252/az2012n2a4 (2012).

2. *At times in the discussion (eg lines 260f, 318f) you veer into rather fuzzy generalisation, whereas the detailed region-by-region discussions are much more effective. On that note, however, the treatment of the West Baltic in your discussion seems rather circumscribed, and could be expanded a little, space permitting.*

The two points raised by the reviewer refer to the association between dairying and cattle and the delay in agricultural adoption in the UK, which are both well supported in the

literature by specific studies. We use these to support the interpretation of our data, namely that dairying is increased in northerly regions where cattle are more frequent and where cereal agriculture was harder to establish.

We have expanded the discussion of the Baltic by inserting the following paragraph, this also clarifies the chronology dealing with the previous point raised by the referee.

Along the western Baltic coastlines of northern Germany, Denmark and southern Sweden, the first evidence for domesticated animals and plants appears at ca. 4,000 cal BC associated with the emergence of the Early Neolithic Funnel Beaker Culture (TRB)^{e.g. 60}. At the same time there is a change in material culture, notably from Late Mesolithic Ertebølle to TRB pottery, but unlike other areas of Europe the exploitation of wild terrestrial game and fishing continued to be economically significant¹⁶. Nevertheless, even at these coastal sites, dairy products feature among the commodities present in these earliest TRB ceramics. They were often processed separately, particularly in small beakers, flasks and bowls⁶¹. The variable use of pottery at coastal TRB sites encompassing both aquatic, dairy and other terrestrial resources may well be a consequence of the interaction of farmers and indigenous foragers. Further genomic analysis is needed to clarify the nature of such interactions but, here at least, indigenous and well-established culinary practices seemed to have persisted well into the Neolithic.

(lines 314-326)

3. *There is an inherent limitation in the genre of papers for Nature and some other leading scientific journals, in that there is space for condensed presentation of large datasets and their analysis, but less room for discussion of complex issues of interpretation and implications. Your paper illustrates this rather well. You refer in the closing discussion to a possible range of explanatory factors (eg line 320 'different economic and cultural traditions') but this could involve all manner of things, and one could wish for a much more detailed climate record against which to compare your data, as well as, in an ideal/non-space restricted world, more detail on individual archaeological contexts as opposed to the broad regionalised pictures given. The way through this if you are redrafting parts of the paper may simply be to acknowledge the complexity of the many issues raised, and to signal the need for ongoing detailed discussion elsewhere.*

Of course, we agree with the reviewer on this point. However, on the following line we do go on to qualify many of these aspects. At the end of the paper we also briefly mention some factors that may have influenced pottery use. However, as the referee is well aware, the drivers that lead the spread of agriculture and livestock in Europe is an on-going debate. Many factors have been proposed for different areas, since the beginning of the research based on Neolithic. Overall, we think a key message from this paper is to

emphasise the regional response to the arrival of agriculture rather than the grand one-size-fits-all narrative as often proffered in short Nature papers.

We have rephrased the sentence (lines 328-333)

“Overall, our study shows that when the ‘Neolithic’ arrived at different regions along the Atlantic coast there were different regional responses. We suggest these responses were influenced both by the different economic and cultural traditions of the farmers who migrated to these new territories, the environments they moved into, and the response of local foragers. Economic adaptations were needed before higher latitudes could be used for food production, resulting in hiatuses in the Neolithic expansion.”

4. ***Line 315: I would rephrase 'cereal agriculture failed' as there are complex issues surrounding this question and the trajectory may not be the same everywhere across Britain and Ireland.***

We have rephrased the sentence and have cited an additional reference as suggested by ref 3:

“Indeed, the available archaeobotanical evidence shows that the degree of arable farming in Britain varied considerably both locally and regionally during the Neolithic period⁵⁸, and in some regions it may even have failed following its initial introduction⁵⁹. (lines 309-312)

References

Bishop, R. R. Did Late Neolithic farming fail or flourish? A Scottish perspective on the evidence for Late Neolithic arable cultivation in the British Isles. *World Archaeol.* **47**, 834–855 (2015)

5. ***Typos/syntax:***

Line 121 needs better punctuation

Line 262 'transects' = transect

Line 288 'forager' = foragers

Line 335 should/could read 'can be inferred'

Line 337 'influence' = influenced

All typos and syntax mistakes have been included in the resubmission version of our paper.

Response to reviewer 3:

1. *A minor quibble is that the section setting out of the research context underplays the body of research on the transition along the Atlantic façade, including those addressing the importance of dairying from a faunal perspective (see especially the work of Anne Tresset). Human DNA is mentioned on a number of occasions, but there is also a considerable body of cattle DNA for the region that is relevant for setting the background scene.*

We thank the reviewer for the suggestion concerning the works of Balasse, Tresset and colleagues. The references relate to the later (middle Neolithic) site of Bercy in Northern France but nonetheless we believe this is relevant to our discussion regarding the introduction of dairying to Britain and how some of the environmental constraints could have been mitigated leading to greater dairy intensification. We have added the following and cited the papers that the referee suggests and an additional one and we have rephrased the paragraph (lines 301-314):

The reproductive timing of both cattle and sheep would have been further environmentally constrained⁵⁴, requiring cultural and biological adaptation to suit local conditions. Isotope analysis of cattle from Bercy in Northern France, shows that the extension of their birthing season and their early weaning to suit more intensified dairying was established at least by the start of the 4th millennium cal BC⁵⁵⁻⁵⁶, corresponding to the arrival of cattle into Britain. Milk and dairy products brought nutritional benefits, such as sources of fat and vitamin D. The latter may have been particularly critical populations moving to higher latitudes where less of this vitamin is produced in vivo due to the reduced exposure to sunlight⁵⁷. Dairying may also have been particularly important to farming populations struggling to establish cereal agriculture as they expanded into new territories sub-optimal for this purpose. Indeed, the available archaeobotanical evidence shows that the degree of arable farming in Britain varied considerably both locally and regionally during the Neolithic period⁵⁸, and in some regions it may even have failed following its initial introduction⁵⁹. (lines 299-312)

References

Balasse, M., Boury, L., Ughetto-Monfrin, J. & Tresset, A. Stable isotope insights (δ 18O, δ 13C) into cattle and sheep husbandry at Bercy (Paris, France, 4th millennium BC): Birth seasonality and winter leaf foddering. *Environ. Archaeol.* **17**, 29–44, doi: 10.1179/1461410312Z.0000000003 (2012).

Balasse, M. & Tresset, A. Early Weaning of Neolithic Domestic Cattle (Bercy, France) Revealed by Intra-tooth Variation in Nitrogen Isotope Ratios. *Journal of Archaeological Science* **29**, 853–859, doi: 10.1006/JASC.2001.0725 (2002).

Balasse, M. & Tresset, A. Environmental constraints on the reproductive activity of domestic sheep and cattle: what latitude for the herder. *Anthropozoologica* **42**, 71–88 (2007)

2. *The statement that the Meso-Neolithic transition on the Atlantic coast has received relatively little attention (isotopically?) downplays a considerable body of stable isotopic, faunal and material culture research on this question (Arias, Dupont, Kador, Marchand, Montgomery, Richards, Scarre, Schulting, Sheridan, Smits, Tresset, Woodman, etc.).*

We meant in terms of organic residue analysis, following from the previous paragraph. To clarify we have rephrased the sentence “One region that has received relatively little attention **with respect to organic residue analysis** is the Atlantic coast of Europe” (Lines 100-101)

3. *122-4/ the evidence for isotopic evidence from the ‘Northern Isles’ of Britain for the continued exploitation of marine foods is questionable. Two references are cited in support. The Montgomery et al. paper refers to only sporadic (this word even features in the paper’s title) use probably during periods of crop failure in a very marginal environment for farming, while the Charlton et al. paper refers to a culturally entirely Mesolithic site (albeit a very late one, probably overlapping with Neolithic sites elsewhere in Britain) with no evidence of domesticated plants or animals or of Neolithic material culture (e.g., pottery). Also the site is not in the Northern Isles.*

We accept this point and we agree with the referee that there is little stable isotope evidence for marine consumption in Early Neolithic Britain, France or Iberia. To be fair this is very clearly stated in the paragraph earlier on. However, we have removed the Charlton et al. paper and used the word sporadic to refer to the Montgomery paper adding that this was only at most a supplement to a largely terrestrial diet.

Despite this, human remains from this period are extremely scarce in the coastal regions of the Iberian Peninsula and France, and this approach lacks the resolution to rule out marine foods entirely²⁴. There is isotopic evidence from an Early Neolithic site in the Northern Isles of Britain for sporadic consumption of marine resources to supplement diets based largely on domesticated plants and animals²⁵. (lines 114-118)

4. *197/ the probable geographical/environmental difference underlying the results for Portugal and Spain in Fig. 2 might be emphasised by specifying these as ‘central/southern Portugal’ and ‘northern Spain’ since from Figure 1 this is where the samples from these countries originated.*

In light of the reviewer's suggestion we have modified Figures 2 and 3 captions as well as the text in the supplementary information pertaining to these assignments. Thus, we have changed Portugal to “Central-Southern Portugal” and Spain to “Northern Spain”. We have also modified the “Baltic margin” (line 60) to the western Baltic in the abstract.

5. Fig. 2F, can comment on why a small number of samples fall so far outside of the modern reference ellipse for marine oils, in directions that do not seem to follow any mixing line between this and the other reference materials?

Some of the samples plotting outside of the ellipses (especially those with elevated C₁₆ and C₁₈ values) could have been used to process a mixture of resources, including marine oils and ruminant adipose or dairy fats. The mixing line between these is not linear, see for example Craig et al. 2011, PNAS. Specifically, one vessel with a C_{16:0} value of -12.0 and a C_{18:0} value of -23.9 had been used to process/store beeswax (see Heron et al. 2007). In light of this comment, we have inserted the following in the caption of Fig. 2.:

“Note that several of the vessels from the western Baltic fall outside of the ellipses and as such a mixture of resources, including marine oils with ruminant adipose/dairy fats is envisaged”.

6. 237/ Why does the Bayesian modelling in Fig. 3 not include the Baltic area?

The distribution of fatty acid isotope values from the western Baltic associated are not normally distributed around the mean due to mixing with marine oils. Therefore, using a median value for the model would not be representative of the sample.

In the methods section we mention:

“The model was applied to the median isotope values from each region (Supplementary Table 2), excluding the western Baltic where the data are bimodally distributed between marine and terrestrial values.” (lines 219-221)

7. 310/ ungrammatical sentence beginning ‘The latter...’]

We have altered the sentence to the following for clarity:

“The latter may have been particularly critical populations moving to higher latitudes where less of this vitamin is produced in vivo due to the reduced exposure to sunlight⁵⁷”.
(lines 305-307)

8. 315/ Might acknowledge that there is still debate concerning the putative failure of arable farming in EN Britain based on the scarcity of cereals in archaeobotanical assemblages (cf. Bishop 2015).

We now acknowledge the debate regarding the report that Neolithic agriculture failed in Britain. We have modified the sentence to highlight the regional variability as pointed out in Bishop 2015. Also see response to reviewer 2.

“Indeed, the available archaeobotanical evidence shows that the degree of arable farming in Britain varied considerably both locally and regionally during the Neolithic period⁵⁸, and in some regions it may even have failed following its initial introduction⁵⁹.” (lines 309-312).

Reviewers' Comments:

Reviewer #1:

Remarks to the Author:

Answers to my review (Reviewer 1)

Point 1 – Novelty

Line 100 – “One region that has received relatively little attention with respect to organic residue analysis is the Atlantic coast of Europe”. The authors ought to cite the existing studies that have been carried out in the S England / N England / Ireland / Scotland, which represent 413 potsherds with $\delta^{13}\text{C}$ values (out of the 647 presented in this paper), that is to say almost 64% of the dataset.

Point 2 - Number of sherds

Thank you for amending that point.

Point 3 - Cattle-based agriculture

Supp Table 3 only collate the faunal data from a subset of the sites studied in this manuscript (ie the unpublished data). Presenting the whole dataset would be useful (but might be a lot of work! – so at the editor’s discretion).

Point 4

There remains considerable important problems with the representation of data in Figure 1. This is of importance since the figure is misleading and yet relied upon heavily to support the main findings of the paper.

One problem previously raised at first review is that it is unjustified to extrapolate beyond the geographic coverage of the data. Unfortunately, the authors’ attempt to resolve this has left the figure even more misleading. Fig 1D is based on the precision of individual $\delta^{13}\text{C}$ values of samples. Whilst these contribute some uncertainty to the $\delta^{13}\text{C}$ values estimates at each site, this becomes trivial when calculating averages across a reasonable sample size, and the effect of intra-site variation and small sample sizes provides a far greater source of uncertainty. But even this misses the point. The reason it is unjustified to extend the interpolated surface to extrapolate beyond the geographic range of the data is because there are spatial limits on the sphere of influence a site has, and the uncertainty on the inferred (interpolated) surface grows hugely at locations that are far from a data point. There are statistical methods to estimate this, for example cross-validation can be used to estimate a sensible bandwidth, which considers the predictive success of the interpolated surface based not only on the sample size but also allows sites to have a greater influence if there is good spatial structure or autocorrelation in the data. In contrast, the standard error of the $\delta^{13}\text{C}$ values measurements is utterly irrelevant to the issue raised, and it is beyond my imagination why an interpolated map of this would provide any value to the reader beyond deliberately misleading them. Similarly, the authors also misrepresent the chronological uncertainty with Fig 1B. The uncertainty around the true date of the site is of importance (the range of uncertainty in Supplementary table 1 is typically c. 500 years (sometimes 1,500 yrs) but Fig 1B suggests the uncertainty is typically c.100. In fact it seems this interpolation is using the average spread of individual calibrated dates. This is derived from the precision of the radiocarbon measurement, which utterly irrelevant to the chronological uncertainty of the sites. Again this serves merely to mislead.

For both the chronology (Fig 1A, 1B) and milk proxy (Fig 1C, 1D), the lack explanation of the interpolation methods used renders it difficult to even vaguely replicate this work.

Given the substantial inter-site variation in the data, the smoothness of the interpolated surface appears implausible and suggests the interpolation algorithm has been mis-specified. Therefore, it is important that the data values at each site are also represented. This can be easily achieved by colouring the sites appropriately (either utilising the same legend colour scheme, perhaps with a black

ring to avoid invisibility if they match the background colour; or with an alternative colour scheme). In addition: The authors must explain how they have accounted for vastly different sample sizes for the milk proxy $\delta^{13}\text{C}$ values C (Fig 1C and 1D). For example, many sites have % dairy fat estimates of either 100% or 0 % based on just a single sample. Therefore it would be statistically unjustified to apply interpolation with equal weighting to each site.

Point 5 – LP
OK

Others aspects

My first review was short as I was advising the editorial board to reject the manuscript. I have thus few more comments on the corrected manuscript that should help improve the manuscript of publication in Nature Comms.

Fig. 2 – Reference animal fats $\delta^{13}\text{C}$ values are cited to come from ref 40 (Lucquin et al. 2018). This paper does not seem to have similar ellipses and I would suggest that the authors cite the primary literature on modern reference fats.

Reviewer 2, point 2 – The authors mention “beakers, flasks and bowls” citing ref 41 – this seems appropriate as the works “flasks and bowls” do not even appear in the paper.

l. 184 – references on $\delta^{13}\text{C}$ values for modern reference fats used in the study to be added.

Reviewer 3, point 5 (l. 194) – The authors are reported $\delta^{13}\text{C}$ values for a sherd that they describe as a mixture of animal fats and beeswax. However, beeswax contain C16:0 fatty acids, and thus the $\delta^{13}\text{C}_{16:0}$ value they quote reflect that of the animal fat but also that of beeswax. It is thus highly inappropriate to report $\delta^{13}\text{C}$ values of mixtures of animal fats if they are mixed with (significant amount of) beeswax.

l. 214 – Why not citing ref 18 here for fish remains in the Atlantic archipelagos (ie a big part of the study of interest)?

l. 361 – The authors should cite Correa Ascencio and Evershed 2014 as this paper proposes a rigorous testing of the direct methanolic extraction.

Fig. 2 – The figure captions should cite the papers from which the reference dataset and most of the datapoints are derived, so that the caption becomes standalone.

Supp Source data – Replace “elaboracion” by “creation” in the caption.

Supp methods – Cite the primary literature for the extraction of lipids from archaeological pottery.

Cite Correa Ascencio and Evershed 2015 for the extraction (see above). N,O in italics for the BSTFA. Subscript “-1” in the velocity (46.57cm.s⁻¹). Cite the primary literature for the use of GC-C-IRMS to distinguish animal fat types. Cite Rieley 1994 for the $\delta^{13}\text{C}$ correction following the addition of carbon during methylation.

Supp Table 1 – when the authors are reporting published data, it would be worth mentioning the reference of the original publication (that is done in the Additional Dataset 2). Change “Recovery” to “Recovered”.

Supp Table 2 – again, report published, unpublished data and respective relevant literature.

Supp Table 3 – rather incomplete as it does not show all the sites cited in the study (ie published data).

Reviewer #2:

Remarks to the Author:

For this resubmission I am only reacting specifically to the responses to my previous comments as Referee 2. You have taken account of all four substantive points raised by me, and have adjusted your wording and referencing satisfactorily. It is for the other two referees to respond in detail to the other

responses to their comments, but in general terms these also look to have been carefully done. I am content therefore that improvements have been made and I stand by my previous recommendation that this paper be accepted for publication. I think it is a significant and useful statement which will be widely cited.

Reviewer #3:

Remarks to the Author:

I feel that the authors have adequately addressed the reviewers' comments. They have provided clear responses to each comments from all three reviewers, making reasoned cases where they have disagreed.

Dear editor,

We are grateful for considering our manuscript entitled "*Latitudinal gradient in dairy production with the introduction of farming in Atlantic Europe*" for publication in *Nature Communications*. In our reviewed version, we have included some modifications related to the comments and suggestions made by the reviewers. In addition, we would like to include some comments additional to the general concerns addressed by the reviewers.

Complete Reviewers' comments:

Reviewer #1 (Remarks to the Author):

Answers to my review (Reviewer 1)

Point 1 – Novelty

Line 100 – “One region that has received relatively little attention with respect to organic residue analysis is the Atlantic coast of Europe”. The authors ought to cite the existing studies that have been carried out in the S England / N England / Ireland / Scotland, which represent 413 potsherds with $\delta^{13}\text{C}$ values (out of the 647 presented in this paper), that is to say almost 64% of the dataset.

Point 2 - Number of sherds. Thank you for amending that point.

Point 3 - Cattle-based agriculture. Supp Table 3 only collates the faunal data from a subset of the sites studied in this manuscript (i.e. the unpublished data). Presenting the whole dataset would be useful (but might be a lot of work! – so at the editor’s discretion).

Point 4. There remains considerable important problems with the representation of data in Figure 1. This is of importance since the figure is misleading and yet relied upon heavily to support the main findings of the paper. One problem previously raised at first review is that it is unjustified to extrapolate beyond the geographic coverage of the data. Unfortunately, the authors’ attempt to resolve this has left the figure even more misleading. Fig 1D is based on the precision of individual $\delta^{13}\text{C}$ values of samples. Whilst these contribute some uncertainty to the $\delta^{13}\text{C}$ values estimates at each site, this becomes trivial when calculating averages across a reasonable sample size, and the effect of intra-site variation and small sample sizes provides a far greater source of uncertainty. But even this misses the point. The reason it is unjustified to extend the interpolated surface to extrapolate beyond the geographic range of the data is because there are spatial limits on the sphere of influence a site has, and the uncertainty on the inferred (interpolated) surface grows hugely at locations that are far from a data point. There are statistical methods to estimate this, for example cross-validation can be used to estimate a sensible bandwidth, which considers the predictive success of the interpolated surface based not only on the sample size but also allows sites to have a greater influence if there is good spatial structure or autocorrelation in the data. In contrast, the standard error of the $\delta^{13}\text{C}$ values measurements is utterly irrelevant to the issue raised, and it is beyond my imagination why an interpolated map of this would provide any value to the reader beyond deliberately misleading them.

Similarly, the authors also misrepresent the chronological uncertainty with Fig 1B. The uncertainty around the true date of the site is of importance (the range of uncertainty in Supplementary table 1 is typically c. 500 years (sometimes 1,500 yrs) but Fig 1B suggests the uncertainty is typically c.100. In fact it seems this interpolation is using the average spread of individual calibrated dates. This is derived from the precision of the radiocarbon measurement,

which utterly irrelevant to the chronological uncertainty of the sites. Again this serves merely to mislead.

For both the chronology (Fig 1A, 1B) and milk proxy (Fig 1C, 1D), the lack explanation of the interpolation methods used renders it difficult to even vaguely replicate this work. Given the substantial inter-site variation in the data, the smoothness of the interpolated surface appears implausible and suggests the interpolation algorithm has been mis-specified. Therefore, it is important that the data values at each site are also represented. This can be easily achieved by colouring the sites appropriately (either utilising the same legend colour scheme, perhaps with a black ring to avoid invisibility if they match the background colour; or with an alternative colour scheme).

In addition: The authors must explain how they have accounted for vastly different sample sizes for the milk proxy $\delta^{13}\text{C}$ values C (Fig 1C and 1D). For example, many sites have % dairy fat estimates of either 100% or 0 % based on just a single sample. Therefore it would be statistically unjustified to apply interpolation with equal weighting to each site.

Point 5 – LP. OK

Others aspects

My first review was short as I was advising the editorial board to reject the manuscript. I have thus few more comments on the corrected manuscript that should help improve the manuscript of publication in Nature Comms.

Fig. 2 – Reference animal fats $\delta^{13}\text{C}$ values are cited to come from ref 40 (Lucquin et al. 2018). This paper does not seem to have similar ellipses and I would suggest that the authors cite the primary literature on modern reference fats.

Reviewer 2, point 2 – The authors mention “beakers, flasks and bowls” citing ref 41 – this seems appropriate as the works “flasks and bowls” do not even appear in the paper.

l. 184 – references on $\delta^{13}\text{C}$ values for modern reference fats used in the study to be added.
Reviewer 3, point 5 (l. 194) – The authors are reported $\delta^{13}\text{C}$ values for a sherd that they describe as a mixture of animal fats and beeswax. However, beeswax contain C16:0 fatty acids, and thus the $\delta^{13}\text{C}_{16:0}$ value they quote reflect that of the animal fat but also that of beeswax. It is thus highly inappropriate to report $\delta^{13}\text{C}$ values of mixtures of animal fats if they are mixed with (significant amount of) beeswax.

l. 214 – Why not citing ref 18 here for fish remains in the Atlantic archipelagos (ie a big part of the study of interest)?

l. 361 – The authors should cite Correa Ascencio and Evershed 2014 as this paper proposes a rigorous testing of the direct methanolic extraction.

Fig. 2 – The figure captions should cite the papers from which the reference dataset and most of the datapoints are derived, so that the caption becomes standalone.

Supp Source data – Replace “elaboracion” by “creation” in the caption.

Supp methods – Cite the primary literature for the extraction of lipids from archaeological pottery. Cite Correa Ascencio and Evershed 2015 for the extraction (see above). N,O in italics for the BSTFA. Subscript “-1” in the velocity (46.57cm.s-1).

Cite the primary literature for the use of GC-C-IRMS to distinguish animal fat types.

Cite Rieley 1994 for the $\delta^{13}\text{C}$ correction following the addition of carbon during methylation.

Supp Table 1 – when the authors are reporting published data, it would be worth mentioning the reference of the original publication (that is done in the Additional Dataset 2).

Change “Recovery” to “Recovered”.

Supp Table 2 – again, report published, unpublished data and respective relevant literature.

Supp Table 3 – rather incomplete as it does not show all the sites cited in the study (ie published data).

Reviewer #2 (Remarks to the Author):

For this resubmission I am only reacting specifically to the responses to my previous comments as Referee 2. You have taken account of all four substantive points raised by me, and have adjusted your wording and referencing satisfactorily. It is for the other two referees to respond in detail to the other responses to their comments, but in general terms these also look to have been carefully done. I am content therefore that improvements have been made and I stand by my previous recommendation that this paper be accepted for publication. I think it is a significant and useful statement which will be widely cited.

Reviewer #3 (Remarks to the Author):

I feel that the authors have adequately addressed the reviewers' comments. They have a provided clear responses to each comments from all three reviewers, making reasoned cases where they have disagreed.

Response to reviewer 1:

Point 1 – Novelty.

Line 100 – “One region that has received relatively little attention with respect to organic residue analysis is the Atlantic coast of Europe”. The authors ought to cite the existing studies that have been carried out in the S England / N England / Ireland / Scotland, which represent 413 potsherds with $\delta^{13}\text{C}$ values (out of the 647 presented in this paper), that is to say almost 64% of the dataset.

We have modified the sentence:

“One region that has received relatively little attention with respect to organic residue analysis is the Atlantic coast of Europe, with studies so far confined to Britain and Ireland^{18, 20}.”

Point 2 - Number of sherds. Thank you for amending that point.

Point 3 - Cattle-based agriculture. Supp Table 3 only collates the faunal data from a subset of the sites studied in this manuscript (i.e. the unpublished data). Presenting the whole dataset would be useful (but might be a lot of work! – so at the editor’s discretion).

We have included information about the number of faunal remains, including NISP values, when available for each of the sites we have obtained pottery from

(Supplementary Table 3). We agree that presenting the whole database for the Atlantic coast of Europe could be useful but it is not the goal of this article. The data we present together with the references provided to regional overviews of Early Neolithic faunal assemblages is more than adequate to support our points that (a) faunal preservation is generally poor, and (b) that cattle are more frequent in the north.

Point 4. There remains considerable important problems with the representation of data in Figure 1. This is of importance since the figure is misleading and yet relied upon heavily to support the main findings of the paper.

One problem previously raised at first review is that it is unjustified to extrapolate beyond the geographic coverage of the data. Unfortunately, the authors' attempt to resolve this has left the figure even more misleading. Fig 1D is based on the precision of individual $\delta^{13}C$ values of samples. Whilst these contribute some uncertainty to the $\delta^{13}C$ values estimates at each site, this becomes trivial when calculating averages across a reasonable sample size, and the effect of intra-site variation and small sample sizes provides a far greater source of uncertainty. But even this misses the point. The reason it is unjustified to extend the interpolated surface to extrapolate beyond the geographic range of the data is because there are spatial limits on the sphere of influence a site has, and the uncertainty on the inferred (interpolated) surface grows hugely at locations that are far from a data point. There are statistical methods to estimate this, for example cross-validation can be used to estimate a sensible bandwidth, which considers the predictive success of the interpolated surface based not only on the sample size but also allows sites to have a greater influence if there is good spatial structure or autocorrelation in the data. In contrast, the standard error of the $\delta^{13}C$ values measurements is utterly irrelevant to the issue raised, and it is beyond my imagination why an interpolated map of this would provide any value to the reader beyond deliberately misleading them.

Similarly, the authors also misrepresent the chronological uncertainty with Fig 1B. The uncertainty around the true date of the site is of importance (the range of uncertainty in Supplementary table 1 is typically c. 500 years (sometimes 1,500 yrs) but Fig 1B suggests the uncertainty is typically c.100. In fact it seems this interpolation is using the average spread of individual calibrated dates. This is derived from the precision of the radiocarbon measurement, which utterly irrelevant to the chronological uncertainty of the sites. Again this serves merely to mislead. For both the chronology (Fig 1A, 1B) and milk proxy (Fig 1C, 1D), the lack explanation of the interpolation methods used renders it difficult to even vaguely replicate this work. Given the substantial inter-site variation in the data, the smoothness of the interpolated surface appears implausible and suggests the interpolation algorithm has been mis-specified. Therefore, it is important that the data values at each site are also represented. This can be easily achieved by colouring the sites appropriately (either utilising the same legend colour scheme, perhaps with a black ring to avoid invisibility if they match the background colour; or with an alternative colour scheme). In addition: The authors must explain how they have accounted for vastly different sample sizes for the milk proxy $\delta^{13}C$ values C (Fig 1C and 1D). For example, many sites have % dairy fat estimates of either 100% or 0 % based on just a single sample. Therefore it would be statistically unjustified to apply interpolation with equal weighting to each site.

We have addressed and discussed this point directly with the editorial team highlighting the misunderstanding the reviewer has made as well as responding to their valid comments. Hopefully have reached a compromise by adding extra data to Fig 1 and clarifying the interpolation approach, which we argue has been suitably applied.

Figure: We have tried many different iterations of Fig 1, including adding extra panels and adding the mean and standard deviations to each site location as we initially suggested. However, given the number of sites (n=62), adding this amount of information renders the figure illegible. Instead we have added the regional mean and standard deviations to the figure so the reader can distinguish ‘actual data’ from the interpolated map. We have added numbers to the 62 site locations which can then be used to look up the mean and standard deviation for the isotope values (Supplementary Data File 3) which we provide in the supplementary data as well as the chronological information.

Interpolation Method: We have greatly expanded the description of the interpolation method in the main body of the text, inserted appropriate references and commented on the associated uncertainty. This should allow our methodology to be followed and should leave the reader in possession of the full facts regarding its suitability.

Point 5 – LP. OK

Others aspects

My first review was short as I was advising the editorial board to reject the manuscript. I have thus few more comments on the corrected manuscript that should help improve the manuscript of publication in Nature Comms.

Fig. 2 – Reference animal fats $\delta^{13}\text{C}$ values are cited to come from ref 40 (Lucquin et al. 2018). This paper does not seem to have similar ellipses and I would suggest that the authors cite the primary literature on modern reference fats.

This reference does not appear in Figure 2. Caption of Figure 2 is:

Fig. 2. $\delta^{13}\text{C}$ values of $\text{C}_{16:0}$ and $\text{C}_{18:0}$ *n*-alkanoic acids on Early Neolithic pottery from the Atlantic coast of Europe and the western Baltic ($n = 647$). **A.** Central-Southern Portugal. **B.** Northern Spain. **C.** France and the Channel Islands. **D.** Southern England. **E.** Northern England, Ireland and Scotland. **F.** Western Baltic. The 68% confidence ellipses are based on modern European authentic reference fats and oils (ruminant adipose fats, ruminant dairy fats, porcine adipose fats and marine oils) (Additional Data File 3). Note that several of the vessels from the western Baltic fall outside of the ellipses and as such a mixture of resources, including marine oils with ruminant adipose/dairy fats is envisaged.

Summary of $\delta^{13}\text{C}$ values of $\text{C}_{16:0}$ and $\text{C}_{18:0}$ *n*-alkanoic acids obtained from modern European authentic reference fats are presented in Supplementary Table 2 where primary literature is cited:

Craig, O. E. *et al.* Ancient lipids reveal continuity in culinary practices across the transition to agriculture in Northern Europe. *Proc. Natl. Acad. Sci. USA* **108**, 17910-17915, doi:10.1073/pnas.1107202108 (2011).

Cramp, L. J. E. *et al.* Immediate replacement of fishing with dairying by the earliest farmers of the northeast Atlantic archipelagos. *Proc. R. Soc. B* **281**, 20132372, doi: 10.1098/rspb.2013.2372 (2014).

Dudd, S. N. *Molecular and isotopic characterisation of animal fats in archaeological pottery* (Unpublished PhD, University of Bristol, 1999).

Spangenberg, J. E., Jacomet, S. & Schibler, J. r. Chemical analyses of organic residues in archaeological pottery from Arbon Bleiche 3, Switzerland: evidence for dairying in the late Neolithic. *J. Archaeol. Sci.* **33**, 1-13, doi: 10.1016/j.jas.2005.05.013 (2006).

Bell, J. G. *et al.* Discrimination of Wild and Cultured European Sea Bass (*Dicentrarchus labrax*) Using Chemical and Isotopic Analyses. *J. Agric. Food Chem.* **55**, 5934-5941, doi:10.1021/jf0704561 (2007).

Spiteri, C. D. *Pottery use at the transition to agriculture in the western Mediterranean. Evidence from biomolecular and isotopic characterisation of organic residues in Impressed/Cardial ware vessels* (Unpublished PhD, University of York, 2012).

Recio, C., Martín, Q. & Raposo, C. GC-C-IRMS analysis of FAMES as a tool to ascertain the diet of Iberian pigs used for the production of pork products with high added value. *Grasas y Aceites* **64**, 181-190, doi: 10.3989/gya.130712 (2013).

Carrer, F. *et al.* Chemical Analysis of Pottery Demonstrates Prehistoric Origin for High-Altitude Alpine Dairying. *PLOS One* **11**, e0151442, doi: 10.1371/journal.pone.0151442 (2016).

Reviewer 2, point 2 – The authors mention “beakers, flasks and bowls” citing ref 41 – this seems appropriate as the works “flasks and bowls” do not even appear in the paper.

The finding of dairy products associated with TRB flasks and bowls is explicitly discussed in reference 61. We have changed the sentence to make this clear:

“Previous studies suggest that were often processed separately, particularly in small beakers, flasks and bowls⁶¹.”

l. 184 – references on $\delta^{13}C$ values for modern reference fats used in the study to be added.

Summary of $\delta^{13}C$ values of C_{16:0} and C_{18:0} *n*-alkanoic acids obtained from modern European authentic reference fats are presented in Additional Table 3 where the primary literature is cited, as we mentioned in our previous reply. This Supplementary Table 2 presents the mean, standard deviation and number of items considered by food resource

(porcine adipose fats, ruminant adipose fats, ruminant dairy fats and marine oils). This data has been used in the elaboration of Figure 2 and the Bayesian model.

Reviewer 3, point 5 (l. 194) – The authors are reported $\delta^{13}C$ values for a sherd that they describe as a mixture of animal fats and beeswax. However, beeswax contain C16:0 fatty acids, and thus the $\delta^{13}C_{16:0}$ value they quote reflect that of the animal fat but also that of beeswax. It is thus highly inappropriate to report $\delta^{13}C$ values of mixtures of animal fats if they are mixed with (significant amount of) beeswax.

In the review of the manuscript, we have not found mention to this comment. We have identified traces of palmitic acid esters in some of the sherds however there is no evidence that beeswax made a significant contribution in any case.

l. 214 – Why not citing ref 18 here for fish remains in the Atlantic archipelagos (ie a big part of the study of interest)?

We have cited this article although unfortunately it provides no reference to the primary zooarchaeological datasets. We have also cited reference 25 which discusses the scant evidence for fish and mollusc exploitation in Britain during the Early Neolithic,

l. 361 – The authors should cite Correa Ascencio and Evershed 2014 as this paper proposes a rigorous testing of the direct methanolic extraction.

We thank the referee for this suggestion but we have decided to include the Craig et al. (2013) and Papakosta et al. (2015) references because we are using a different concentration of sulphuric acid during the extraction to the Correo-Ascencio and Evershed protocol. To be more specific, in the Correo-Ascencio and Evershed paper (2014) the solution of H_2SO_4 -MeOH was at 2%(v/v), whilst we used a concentration at 16% (1:5 v/v).

Fig. 2 – The figure captions should cite the papers from which the reference dataset and most of the datapoints are derived, so that the caption becomes standalone.

We thank your comments related to this topic. Following the reviewer's advice, we have included a reference to the Supplementary Data Files in figure caption, where information about isotopic values for authentic reference fats and oils (ruminant adipose fats, ruminant dairy fats, porcine adipose fats and marine oils) is included with the original bibliographical references.

Supp Source data – Replace “elaboracion” by “creation” in the caption.

This modification has been included in the resubmission version of our paper in Source data and supplementary tables captions.

Supp methods – Cite the primary literature for the extraction of lipids from

archaeological pottery. Cite Correa Ascencio and Evershed 2015 for the extraction (see above).

We thank your comments related to this topic and we think that the reviewer's comment referred to the Correa Ascencio and Evershed 2014 paper. As we have pointed out above, we have decided to include the Craig et al. (2013) and Papakosta et al. (2015) references because we are using a different concentration of sulphuric acid during the extraction.

The Supplementary Methods section has been modified and all methods information has been included in the main manuscript, as suggested by the editorial team.

N₂O in italics for the BSTFA. Subscript "-1" in the velocity (46.57cm.s⁻¹).

All typos and syntax mistakes have been included in the resubmission version of our paper.

Cite the primary literature for the use of GC-C-IRMS to distinguish animal fat types. Cite Rieley 1994 for the $\delta^{13}C$ correction following the addition of carbon during methylation.

We have included pertinent and most important references relating to the topic and its application to organic residue analysis in archaeological pottery.

Supp Table 1 – when the authors are reporting published data, it would be worth mentioning the reference of the original publication (that is done in the Additional Dataset

Supplementary table 1 just includes pottery samples directly analysed in this study. To highlight this point, we have modified the caption of the table

“Supplementary Table 1. Summary of pottery samples directly analysed in this study. *Approximate chronology is based on available calibrated radiocarbon dates for each site when no radiocarbon dates were available, most representative vessel typology is highlighted. **Estimated % of samples has been established considering the *n* number of vessels, when available, or the total *n* of pottery sherds recovered in the same archaeological context. ***Lipid concentration was quantified from the acidified methanol extract only. "n.d." indicates that no data are available.”

2). Change “Recovery” to “Recovered”.

All typos and syntax mistakes have been included in the resubmission version of our paper.

Supp Table 2 – again, report published, unpublished data and respective relevant literature.

Supplementary Table 2 presents the median values of *n*-alkanoic acids from each region used in the Bayesian semiparametric mixed model based on the data presented on Supplementary Data File 2. To clarify this aspect, we have modified the caption of Table 2:

Supplementary Table 2. Median stable carbon isotope ($\delta^{13}\text{C}$) values of $\text{C}_{16:0}$ and $\text{C}_{18:0}$ *n*-alkanoic acids from each region used in the Bayesian semiparametric mixed model. These values are based on Stable carbon isotope ($\delta^{13}\text{C}$) values of $\text{C}_{16:0}$ and $\text{C}_{18:0}$ *n*-alkanoic acids obtained on Early Neolithic pottery from the archaeological sites located in the Atlantic coast of Europe and the western Baltic ($n = 647$) grouped by region (Supplementary Data file 2).

In addition, this table has been renamed as Supplementary Table 3.

Supp Table 3 – rather incomplete as it does not show all the sites cited in the study (ie published data).

We had included information about the number of faunal remains, including NISP values, when available for each site we obtained pottery from (now Supplementary Table 4. Presenting the whole database for the Atlantic coast of Europe would be useful but it is not the goal of this article. For that reason, we have included several bibliographical references so that the reader can consult the faunal assemblages available in the geographical area. In addition, this table has been renamed as Supplementary Table 4.

Response to Reviewer #2 (Remarks to the Author):

For this resubmission I am only reacting specifically to the responses to my previous comments as Referee 2. You have taken account of all four substantive points raised by me, and have adjusted your wording and referencing satisfactorily. It is for the other two referees to respond in detail to the other responses to their comments, but in general terms these also look to have been carefully done. I am content therefore that improvements have been made and I stand by my previous recommendation that this paper be accepted for publication. I think it is a significant and useful statement which will be widely cited.

We would like to thank all the comments and suggestions made by Reviewer 2 that, without any doubt, have improved the quality of the original manuscript.

Reviewer #3 (Remarks to the Author):

I feel that the authors have adequately addressed the reviewers' comments. They have a provided clear responses to each comments from all three reviewers, making reasoned cases where they have disagreed.

We would like to thank all the comments and suggestions made by Reviewer 3 that, without any doubt, have improved the quality of the original manuscript.